# SHAPE MATTERS: UNDERSTANDING THE IMPLICIT BIAS OF THE NOISE COVARIANCE

## ABSTRACT

The noise in stochastic gradient descent (SGD) provides a crucial implicit regularization effect for training overparameterized models. Prior theoretical work largely focuses on spherical Gaussian noise, whereas empirical studies demonstrate the phenomenon that parameter-dependent noise — induced by minibatches or label perturbation — is far more effective than Gaussian noise. This paper theoretically characterizes this phenomenon on a quadratically-parameterized model introduced by Vaskevicius et al. and Woodworth et al. We show that in an over-parameterized setting, SGD with label noise recovers the sparse ground-truth with an arbitrary initialization, whereas SGD with Gaussian noise or gradient descent overfits to dense solutions with large norms. Our analysis reveals that parameter-dependent noise introduces a bias towards local minima with smaller noise variance, whereas spherical Gaussian noise does not.

## 1 INTRODUCTION

One central mystery of deep artificial neural networks is their capability to generalize when having far more learnable parameters than training examples Zhang et al. (2016). To add to the mystery, deep nets can also obtain reasonable performance in the absence of any explicit regularization. This has motivated recent work to study the regularization effect due to the optimization (rather than objective function), also known as *implicit bias* or *implicit regularization* Gunasekar et al. (2017; 2018a;b); Soudry et al. (2018); Arora et al. (2019). The implicit bias is induced by and depends on many factors, such as learning rate and batch size Smith et al. (2017); Goyal et al. (2017); Keskar et al. (2016); Li et al. (2019b); Hoffer et al. (2017), initialization and momentum Sutskever et al. (2013), adaptive stepsize Kingma and Ba (2014); Neyshabur et al. (2015); Wilson et al. (2017), batch normalization Ioffe and Szegedy (2015) and dropout Srivastava et al. (2014).

Among these sources of implicit regularization, the SGD noise is believed to be a vital one (LeCun et al., 2012; Keskar et al., 2016). Previous theoretical works (e.g., Li et al. (2019b)) have studied the implicit regularization effect from the *scale of the noise*, which is directly influenced by learning rate and batch size. However, people have empirically observed that the *shape of the noise* also has a strong (if not stronger) implicit bias. For example, prior works show that mini-batch noise or label noise (label smoothing) – noise in the parameter updates from the perturbation of labels in training – is far more effective than adding spherical Gaussian noise (e.g., see (Shallue et al., 2018, Section 4.6) and Szegedy et al. (2016); Wen et al. (2019)). We also confirm this phenomenon in Figure 1 (left). Thus, understanding the implicit bias of the noise shape is crucial. Such an understanding may also apply to distributed training because synthetically adding noise may help generalization if parallelism reduces the amount of mini-batch noise (Shallue et al., 2018).

In this paper, we theoretically study the effect of the shape of the noise, demonstrating that it can provably determine generalization performance at convergence. Our analysis is based on a nonlinear quadratically-parameterized model introduced by (Woodworth et al., 2020; Vaskevicius et al., 2019), which is rich enough to exhibit similar empirical phenomena as deep networks. Indeed, Figure 1 (right) empirically shows that SGD with mini-batch noise or label noise can generalize with arbitrary initialization without explicit regularization, whereas GD or SGD with spherical Gaussian noise cannot. We aim to analyze the implicit bias of label noise and Gaussian noise in the quadratically-parametrized model and explain these empirical observations.

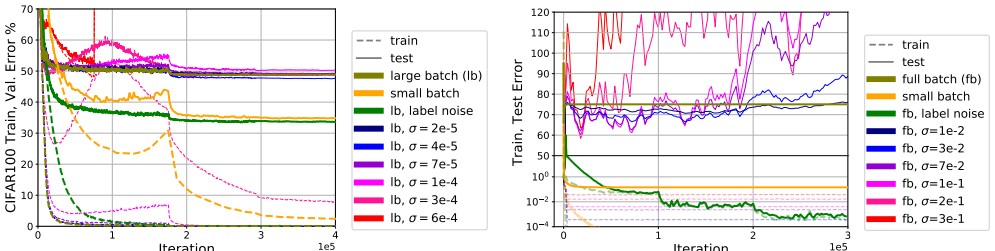

Figure 1: **The effect of noise covariance in neural network and quadratically-parameterized models.** We demonstrate that label noise induces a stronger regularization effect than Gaussian noise. In both real and synthetic data, adding label noise to large batch (or full batch) SGD updates can recover small-batch generalization performance, whereas adding Gaussian noise with optimally-tuned variance $\sigma^2$ cannot. **Left:** Training and validation errors on CIFAR100 for VGG19. Adding Gaussian noise to large batch updates gives little improvement (around 2%), whereas adding label noise recovers the small-batch baseline (around 15% improvement). **Right:** Training and validation error on a 100-dimensional quadratically-parameterized model defined in Section 2. Similar to deep models, label noise or mini-batch noise leads to better solutions than optimally-tuned spherical Gaussian noise. Moreover, Gaussian noise causes the parameter to diverge after sufficient mixing, as suggested by our negative result for Langevin dynamics (Theorem 2.2). More details are in Section A.

We choose to study label noise because it can replicate the regularization effects of minibatch noise in both real and synthetic data (Figure 1), and has been used to regularize large-batch parallel training (Shallue et al., 2018). Moreover, label noise is less sensitive to the initialization and the optimization history than mini-batch noise, which makes it more amenable to theoretical analysis. For example, in an extreme case, if we happen to reach or initialize at a solution that overfits the data exactly, then mini-batch SGD will stay there forever because both the gradient and the noise vanish (Vaswani et al., 2019). In contrast, label noise will not accidentally vanish, so the analysis is more tractable. Understanding label noise may lead to understanding mini-batch noise or replacing it with other more robust choices.

In our setting, we prove that with a proper learning rate schedule, SGD with label noise recovers a sparse ground-truth classifier and generalizes well, whereas SGD with spherical Gaussian noise generalizes poorly. Concretely, SGD with label noise biases the parameter towards the low sparsity regime and exactly recovers the sparse ground-truth, even when the initialization is arbitrarily large (Theorem 2.1). In this same regime, noise-free gradient descent quickly overfits because it trains in the NTK regime (Jacot et al., 2018; Chizat and Bach, 2018). Adding Gaussian noise is insufficient to fix this, as this algorithm would end up sampling from a Gibbs distribution with infinite partition function and fail to converge to the ground-truth (Theorem 2.2). In summary, with not too small learning rate or noise level, label noise suffices to bias the parameter towards sparse solutions without relying on a small initialization, whereas Gaussian noise cannot.

Our analysis suggests that the fundamental difference between label or mini-batch noise and Gaussian noise is that the former is parameter-dependent, and therefore introduces stronger biases than the latter. The conceptual message highlighted by our analysis is that there are *two* possible implicit biases induced by the noise: 1. prior work (Keskar et al., 2016) shows that by escaping sharp local minima, noisy gradient descent biases the parameter towards more robust solutions (i.e, solutions with low curvature, or "flat" minima), and 2. when the noise covariance varies across the parameter space, there is another (potentially stronger) implicit bias effect toward parameters where the noise covariance is smaller. Label or mini-batch noise benefits from both biases, whereas Gaussian noise is independent of the parameter, so it benefits from the first bias but not the second. For the quadratically-parameterized model, this first bias is not sufficient for finding solutions with good generalization because there is a large set of overfitting global minima of the training loss with reasonable curvature. In contrast, the covariance of label noise is proportional to the scale of the parameter, inducing a much stronger bias towards low norm solutions which generalize well.

## 1.1 ADDITIONAL RELATED WORKS

Closely related to our work, Blanc et al. (2019) and Zhu et al. (2019) also theoretically studied implicit regularization effects that arise due to shape, rather than scale, of the noise. However, they only considered the *local* effect of the noise near some local minimum of the loss. In contrast, our

work analyzes the global effect of noise. For a more detailed comparison with (Blanc et al., 2019), see Section 2.2.

Woodworth et al. (2020); Vaskevicius et al. (2019) analyze the effect of initialization for the same model that we study, showing that large initialization trains in the NTK regime (shown to generalize poorly (Wei et al., 2019; Ghorbani et al., 2019)) whereas small initialization does not. We show that when the initialization is large, adding noise helps avoid the NTK regime (Li and Liang, 2018; Jacot et al., 2018; Du et al., 2018b; Woodworth et al., 2020) without explicit regularization.

Several previous works have studied generalization bounds and training dynamics of SGD with state-dependent noises for more general models. Hardt et al. (2015) derived stability-based generalization bounds for mini-batch SGD based on training speed. Cheng et al. (2019) proved that SGD with state-dependent noises has iterate distribution close to the corresponding continuous stochastic differential equation with the same noise covariance. Meng et al. (2020); Xie et al. (2020) showed that SGD with state-dependent noises escapes local minimum faster than SGD with spherical Gaussian noise .

There has been a line of work empirically studying how noise influences generalization. Keskar et al. (2016) argued that large batch training will converge to "sharp" local minima which do not generalize well. Hoffer et al. (2017) argued that large batch size doesn't hurt generalization much if training goes on long enough and additional noise is added with a larger learning rate. Goyal et al. (2017) and Shallue et al. (2018) showed large batch training with proper learning rate and additional label noise can achieve similar generalization as small batch. Wei and Schwab (2019); Chaudhari and Soatto (2018); Yaida (2018) (heuristically) suggested that SGD may encourage solutions with smaller noise covariance. Martin and Mahoney (2018) used random matrix theory to analyze implicit regularization effects of noises. The noise induced by dropout has been shown to change the expected training objective, hence provides a regularization effect (Mianjy et al., 2018; Mianjy and Arora, 2019; Wei et al., 2020; Arora et al., 2020). Wei et al. (2020) showed that there also exists an implicit bias induced by dropout noise.

Langevin dynamics or the closely-related stochastic gradient descent with spherical Gaussian noise has been studied in previous works Welling and Teh (2011); Teh et al. (2016); Raginsky et al. (2017); Zhang et al. (2017); Mou et al. (2017); Roberts et al. (1996); Ge et al. (2015); Negrea et al. (2019); Neelakantan et al. (2015); Mou et al. (2018). In particular, Raginsky et al. (2017) and Li et al. (2019a) provided generalization bounds for SGLD using algorithmic stability.

Several works have theoretically analyzed other types of implicit biases in simplified settings (Soudry et al., 2018; Gunasekar et al., 2018b; Ji and Telgarsky, 2018a). Gunasekar et al. (2017) and Li et al. (2017) showed that gradient descent finds low rank solutions in matrix completion. Gradient descent has also been shown to maximize the margin in linear and homogeneous models (Soudry et al., 2018; Ji and Telgarsky, 2018b; Nacson et al., 2018; Lyu and Li, 2019; Gunasekar et al., 2018a; Nacson et al., 2019; Poggio et al., 2017). Du et al. (2018a) showed that gradient descent implicitly balances the layers of deep homogeneous models. Other works showed that it may not always be possible to characterize implicit biases in terms of norm (Arora et al., 2019; Razin and Cohen, 2020). Gissin et al. (2019) showed that gradient descent dynamics exhibit different implicit biases based on depth. Li et al. (2019b) studied the implicit regularization effect of a large initial learning rate.

Guo et al. (2018) studies the notion of "elimination singularities" in RBF networks, where optimization runs into a regime with small weight and small gradient, therefore the training can be slowed down. Our paper also involves training trajectory with small weight norm, but instead focuses on its influnece on the generalization performance.

## 2 SETUP AND MAIN RESULTS

### 2.1 SETUP AND BACKGROUNDS

**Parameterization.** We focus on the nonlinear model parametrization: $f_v(x) \triangleq \langle v^{\odot 2}, x \rangle$, where $v \in \mathbb{R}^d$ is the parameter of the model, $x \in \mathbb{R}^d$ is the data, and $v^{\odot 2}$ denotes the element-wise square of $v$. Prior works (Woodworth et al., 2020; Vaskevicius et al., 2019; Li et al., 2017) have studied this model because it is an interesting and informative simplification of nonlinear models. As SGD noise exhibits many of the same empirical behaviors in this simplified model as in deep networks,[1] we use

---

[1]In contrast, the implicit bias of noise wouldn't show up in a simpler linear regression model.

this model as a testbed to develop a mathematical understanding of various sources of implicit biases. As shown in Figure 1, both SGD with mini-batch noise and label noise generalize better than GD or SGD with spherical Gaussian noise.

**Data distribution assumptions and overparametrization.** We assume that there exists a ground-truth parameter $v^\star \in \mathbb{R}^d$ that generates the label $y = \langle v^{\star \odot 2}, x \rangle$ given a data point $x$, which is assumed to be generated from $\mathcal{N}(0, \mathcal{I}_{d \times d})$. A dataset $\mathcal{D} = \left\{ (x^{(i)}, y^{(i)}) \right\}_{i=1}^n$ of $n$ i.i.d data points are generated from this distribution. The implicit bias is only needed in an over-parameterized regime, and therefore we assume that $n \ll d$. To make the ground-truth vector information-theoretically recoverable, we assume that the ground-truth vector $v^\star$ is $r$-sparse. Here $r$ is much smaller than $d$, and casual readers can treat it as a constant. Because the element-wise square in the model parameterization is invariant to any sign flip, we assume $v^\star$ is non-negative without loss of generality. For simplicity, we also assume it only takes value in $\{0, 1\}$.[2] We use $S \subset [d]$ with $|S| = r$ to denote the support of $v^\star$ throughout the paper.

We remark that we can recover $v^\star$ by re-parameterizing $u = v^{\odot 2}$ and applying LASSO (Tibshirani, 1996) in the $u$-space when $n \geq \widetilde{O}(r)$, which is minimax optimal (Raskutti et al., 2012). However, the main goal of the paper, similar to several prior works (Woodworth et al., 2020; Vaskevicius et al., 2019; Li et al., 2017), is to prove that the implicit biases of non-convex optimization can recover the ground truth *without explicit regularization* in the over-parameterized regime when $n = \text{poly}(r) \ll d$.[3] We also assume throughout the paper that $n, d$ are larger than some sufficiently large universal constant.

**Loss function.** We use the mean-squared loss denoted by $\ell^{(i)}(v) \triangleq \frac{1}{4} \left( f_v(x^{(i)}) - y^{(i)} \right)^2$ for the $i$-th example. The empirical loss is written as $\mathcal{L}(v) \triangleq \frac{1}{n} \sum_{i=1}^n \ell^{(i)}(v)$.

**Initialization.** We use a large initialization of the form $v^{[0]} = \tau \cdot \mathbb{1}$ where $\mathbb{1}$ denotes the all 1's vector, where we allow $\tau$ to be arbitrarily large (but polynomial in $d$).

---

**Algorithm 1** Stochastic Gradient Descent with Label Noise

---

**Require:** Number of iterations $T$, a sequence of step sizes $\eta^{[0:T]}$, noise level $\delta$, initialization $v^{[0]}$
 1: **for** $t = 0$ to $T - 1$ **do**
 2:     Sample index $i_t \sim [n]$ uniformly and add noise $s_t \sim \{\pm \delta\}$ to $y^{(i_t)}$.
 3:     Let $\tilde{\ell}^{(i_t)}(v) = \frac{1}{4}(f_v(x^{(i_t)}) - y^{(i_t)} - s_t)^2$
 4:     $v^{[t+1]} \leftarrow v^{[t]} - \eta^{[t]} \nabla \tilde{\ell}^{(i_t)}(v^{[t]})$                     ▷ update with label noise

---

**SGD with label noise.** We study SGD with label noise as shown in Algorithm 1. We sample an example, add label noise sampled from $\{\pm \delta\}$ to the label, and apply the gradient update. Computing the gradient, we obtain the update rule written explicitly as:

$$v^{[t+1]} \leftarrow v^{[t]} - \eta^{[t]} \left( (v^{[t] \odot 2} - v^{\star \odot 2})^\top x^{(i_t)} \right) x^{(i_t)} \odot v^{[t]} + \eta^{[t]} s_t x^{(i_t)} \odot v^{[t]}. \tag{1}$$

**Langevin dynamics/diffusion.** We compare SGD with label noise to Langevin dynamics, which adds *spherical* Gaussian noise to gradient descent (Neal et al., 2011):

$$v^{[t+1]} \leftarrow v^{[t]} - \eta \nabla \mathcal{L}(v^{[t]}) + \sqrt{2\eta/\lambda} \cdot \xi, \tag{2}$$

where the noise $\xi \sim \mathcal{N}(0, \mathcal{I}_{d \times d})$ and $\lambda > 0$ controls the scale of noise. Langevin dynamics (LD) or its more computationally-efficient variant, stochastic gradient Langevin dynamics (SGLD), is known to converge to the Gibbs distribution $\mu(v) \propto e^{-\lambda \mathcal{L}(v)}$ under various settings with sufficiently small learning rate (Roberts et al., 1996; Dalalyan, 2017; Bubeck et al., 2018; Raginsky et al., 2017). In our negative result about Langevin dynamics/diffusion, we directly analyze the Gibbs distribution in order to disentangle the convergence and the generalization. In our negative result about Langevin dynamics/diffusion, we directly analyze the Gibbs distribution to disentangle the convergence and the generalization.

**Notations.** Unless otherwise specified, we use $O(\cdot), \Omega(\cdot), \Theta(\cdot)$ to hide absolute multiplicative factors and $\widetilde{O}(\cdot), \widetilde{\Theta}(\cdot), \widetilde{\Omega}(\cdot)$ to hide poly-logarithmic factors in problem parameters such as $d$ and $\tau$. For

---

[2]Our analysis can be straightforwardly extended to $v^\star$ with other non-zero values.

[3]We also remark that it's common to obtain only sub-optimal sample complexity guarantees in the sparsity parameters with non-convex optimization methods (Li et al., 2017; Ge et al., 2016; Vaskevicius et al., 2019; Chi et al., 2019) due to technical limitations.

example, every occurrence of $\widetilde{O}(x)$ is a placeholder for a quantity $f(x)$ that satisfies that for some absolute constants $c_1, c_2 > 0, \forall x, |f(x)| \leq c_1|x| \cdot \log^{c_2}(d\tau)$.

## 2.2 MAIN RESULTS

Our main result can be summarized by the following theorem, which suggests that stochastic gradient descent with label noise can converge to the ground truth despite a potentially large initialization.

**Theorem 2.1.** *In the setting of Section 2.1, given a target error $\epsilon > 0$. Suppose we have $n \geq \widetilde{\Theta}(r^2)$ samples. For any label noise level $\delta \geq \widetilde{\Theta}(\tau^2 d^2)$, we run SGD with label noise (Algorithm 1) with the following learning rate schedule:*

1. *learning rate $\eta_0 = \widetilde{\Theta}(1/\delta)$ for $T_0 = \widetilde{\Theta}(1)$ iterations,*
2. *learning rate $\eta_1 = \widetilde{\Theta}(1/\delta^2)$ for $T_1 = \widetilde{\Theta}(1/\eta_1)$ iterations,*
3. *learning rate $\eta_2 = \widetilde{\Theta}(\epsilon^2/\delta^2)$ for $T_2 = \widetilde{\Theta}(1/\eta_2)$ iterations.*

*Then, with probability at least $0.9$, the final iterate $v^{[T]}$ at time $T = T_0 + T_1 + T_2$ satisfies*

$$\|v^{[T]} - v^\star\|_\infty \leq \epsilon. \tag{3}$$

*Here $\widetilde{\Theta}(\cdot)$ omits poly-logarithmic dependencies on $1/\epsilon$, $d$ and $\tau$.*

In other words, with arbitrarily large initialization scale $\tau$, we can choose large label noise level and the learning rate schedule so that SGD with label noise succeeds in recovering the ground truth. In contrast, when $\tau$ is large, gradient flow without noise trains in the "kernel" regime as shown by (Woodworth et al., 2020; Chizat and Bach, 2018). The solution in this kernel regime minimizes the RKHS distance to initialization, and in our setting equates to finding a zero-error solution with minimum $\|v^{\odot 2} - v^{[0]\odot 2}\|_2$. Such a solution could be arbitrarily far away when initialization scale $\tau$ is large and therefore have poor generalization. Figure 1 (right) confirms GD performs poorly with large initialization whereas SGD with minibatch or label noise works. We outline the proof of Theorem 2.1 in Section 3.

Blanc et al. (2019) also study the implicit bias of the label noise. For our setting, their result implies that when the iterate is near a global minimum for sufficient time, the iterates will *locally* move to the direction that reduces the $\ell_2$-norm of $v$ by a small distance (that is larger than random fluctuation). However, it does not imply the global convergence to a solution with good generalization with large (or any) initialization, which is what we prove in Theorem 2.1.[4] Moreover, our analysis captures the effect of the large noise or large learning rate – we require the ratio between the noise and the gradient, which is captured by the value $\eta\delta^2$, to be sufficiently large. This is consistent with the empirical observation that good generalization requires a sufficiently large learning rate or small batch (Goyal et al., 2017).

On the other hand, the following negative result for Langevin dynamics demonstrates that adding Gaussian noise fails to recover the ground truth even when $v^\star = 0$. This suggests that spherical Gaussian noise does not induce a strong enough implicit bias towards low-norm solutions.

**Theorem 2.2.** *Assume in addition to the setting in Section 2.1 that the ground truth $v^\star = 0$. When $n \leq d/3$, with probability at least $0.9$ over the randomness of the data, for any $\lambda > 0$, the Gibbs distribution is not well-defined because the partition function explodes:*

$$\int_{\mathbb{R}^d} e^{-\lambda\mathcal{L}(v)} dv = \infty. \tag{4}$$

*As a consequence, Langevin diffusion does not converge to a proper stationary distribution.*

Theorem 2.2 helps explain the behavior in Figure 1, where adding Gaussian noise generalizes poorly for both synthetic and real data. In particular, in Figure 1 (right) adding Gaussian noise causes the parameter to diverge for synthetic data, and Theorem 2.2 explains this observation. A priori, the intuition regarding Langevin dynamics is as follows: as $\lambda \to +\infty$, the Gibbs distribution (if it exists) should concentrate on the manifold of global minima with zero loss. The measure on the manifold of

---

[4]It also appears difficult to generalize the local analysis directly to a global analysis because once the iterate leaves the local minimum, all the local tools do not apply anymore, and it's unclear whether the iterate will converge to a new local minimum or getting stuck at some region.

global minima should be decided by the geometry of $\mathcal{L}(\cdot)$, and in particular, the curvature around the global minimum. As $\lambda \to +\infty$, the mass should likely concentrate at the flattest global minimum (according to some measure of flatness), which intuitively is $v^\star = 0$ in this case.

However, our main intuition is that when $n < d$, even though the global minimum at $v^\star$ is the flattest, there are also many bad global minima with only slightly sharper curvatures. The vast volume of bad global minima dominate the flatness of the global minimum at $v^\star = 0$ for any $\lambda$,[5] and hence the partition function blows up and the Gibbs distribution doesn't exist. The proof of Theorem 2.2 can be found in Section F.

## 3 ANALYSIS OVERVIEW OF SGD WITH LABEL NOISE (THEOREM 2.1)

### 3.1 WARM-UP: UPDATES WITH ONLY PARAMETER-DEPENDENT NOISE

Towards building intuition and tools for analyzing the parameter-dependent noise, in this subsection we start by studying an extremely simplified random walk in one dimensional space. The random walk is purely driven by mean-zero noisy updates and does not involve any gradient updates:

$$v \leftarrow v + \eta \xi \cdot v, \text{ where } \xi \sim \{\pm 1\}. \tag{5}$$

Indeed, attentive readers can verify that when dimension $d = 1$, sample size $n = 1$, and $v^\star = 0$, equation (1) degenerates to the above random walk if we omit the gradient update term (second to last term in equation (1)). We compare it with the standard Brownian motion (which is the analog of gradient descent with spherical Gaussian noise under this extreme simplification)

$$v \leftarrow v + \eta \xi, \text{ where } \xi \sim \mathcal{N}(0, 1). \tag{6}$$

We initialize at $v = 1$. We observe that both random walks have mean-zero updates, so the mean is preserved: $\mathbb{E}[v] = 1$. The variances of the two random walks are also both growing because any mean-zero update increases the variance. Moreover, the Brownian motion diverges because it has a Gaussian marginal with variance growing linearly in $t$, and there is no limiting stationary distribution.

However, the parameter-dependent random walk (5) has dramatically different behavior when $\eta < 1$: the random variable $v$ will eventually converge to $v = 0$ with high probability (though the variance grows and the mean remains at 1.). This is because the variance of the noise depends on the scale of $v$. The smaller $v$ is, the smaller the noise variance is, and so the random walk tends to get "trapped" around 0. This claim has the following informal but simple proof that does not strongly rely on the exact form of the noise and can be extended to more general high-dimensional cases.

Consider an increasing concave potential function $\phi : \mathbb{R}_{\geq 0} \to \mathbb{R}_{\geq 0}$ with $\phi'' < 0$ (e.g., $\phi(v) = \sqrt{v}$ works). Note that when $\eta < 1$, the random variable $v$ stays nonnegative. We can show that the expected potential function decreases after any update

$$\mathbb{E}[\phi(v + \eta \xi v)] \approx \mathbb{E}[\phi(v) + \phi'(v)\eta \xi v + \phi''(v)\eta^2 \xi^2 v^2] \qquad \text{(by Taylor expansion)}$$

$$= \mathbb{E}[\phi(v)] + \mathbb{E}[\phi''(v)\eta^2 v^2] < \mathbb{E}[\phi(v)] \qquad \text{(by } \phi''(v) < 0 \text{ and } \mathbb{E}[\xi] = 0.)$$

With more detailed analysis, we can formalize the Taylor expansion and control the decrease of the potential function, and conclude that $\mathbb{E}[\phi(v)]$ converges to zero. Then, by Markov's inequality, with high probability, $\phi(v)$ is tiny and so is $v$.[6]

**From the 1-D case to the high-dimensional case.** In one dimension, the bias is introduced because of the varying scale of noise (i.e., the norm of the covariance). However, in the high dimensional case, the shape of the covariance also matters. For example, if we generalize the random walk (1) to high-dimensions by running $d$ of the random walks in parallel, then we will observe the same phenomenon, but the noise variances in different dimensions are *not identical* — they depend on the current scales of the coordinates. (Precisely, the noise variance for dimension $k$ is $\eta^2 v_k^2$.) However, suppose we instead add noise of the *same variance* to all dimensions. Even if this variance depends on the norm of $v$ (say, $\eta^2 \|v\|_2^2$), the implicit bias will be diminished, as the smaller coordinates will have relatively outsized noise and the larger coordinates will have relatively insufficient noise.

---

[5]In fact, one can show that if this phenomenon happens for some $\lambda > 0$, then it happens for all other $\lambda$.

[6]The same proof strategy fails for the Brownian motion because $v$ is not always nonnegative, and there is no concave potential function over the real that can be bounded from below.

**Outline of the rest of the subsections.** We will give a proof sketch of Theorem 2.1 that consists of three stages. We first show in the initial stage of the training that label noise effectively decreases the parameter on all dimensions, bringing the training from large initialization to a small initialization regime, where better generalization is possible (Section 3.2). Then, we show in Section 3.3 that when the parameter is decently small, with label noise and a *decayed learning rate*, the algorithm will increase the magnitude of those dimensions in support set of $v^\star$, while keep decreasing the norm of the rest of the dimensions. Finally, with one more decay, the algorithm can recover the ground truth.

## 3.2  Stage 0: Label Noise with Large Learning Rate Reduces the Parameter Norm

We first analyze the initial phase where we use a relatively large learning rate. When the initialization is of a decent size, GD quickly *overfits* to a bad global minimum nearest to the initialization. In contrast, we prove that SGD with label noise biases towards the small norm region, for a similar reason as the random walk example with parameter-dependent noise in Section 3.1.

**Theorem 3.1.** *In the setting of Theorem 2.1, recall that we initialize with $v^{[0]} = \tau \cdot \mathbb{1}$. Assume $n \geq \Theta(\log d)$. Suppose we run SGD with label noise with noise level $\delta \geq \widetilde{\Theta}(\tau^2 d^2)$ and learning rate $\eta_0 \in [\widetilde{\Theta}(\tau^2 d^2/\delta^2), \widetilde{\Theta}(1/\delta)]$ for $T_0 = \widetilde{\Theta}(1/(\eta^2\delta^2))$ iterations. Then, with probability at least $0.99$ over the randomness of the algorithm,*

$$\|v^{[T_0]}\|_\infty \leq 1/d. \tag{7}$$

*Moreover, the minimum entry of $v^{[T_0]}$ is bounded below by $\exp(-\widetilde{O}((\eta\delta)^{-1}))$.*

We remark that our requirement of $\eta$ being large is consistent with the empirical observation that a large initial learning rate helps generalization Goyal et al. (2017); Li et al. (2019b). We provide intuitions and a proof sketch of the theorem in the rest of the subsection and defer the full proof to Section B . Our proof is based on the construction of a *concave* potential function $\Phi$ similar to Section 3.1. We will show that, at every step, the noise has a second-order effect on the potential function and decrease the potential function by a quantity on the order of $\eta^2\delta^2$ (omitting the $d$ dependency).[7]  On the other hand, the gradient step may increase the potential by a quantity at most on the order of $\eta$ (omitting $d$ dependency again). Therefore, when $\eta^2\delta^2 \gtrsim \eta$, we expect the algorithm to decrease the potential and the parameter norm.

In particular, we define $\Phi(v) \triangleq \sum_{k=1}^d \phi(v_k) = \sum_{k=1}^d \sqrt{v_k}$. By the update rule 2.1, the update for a coordinate $k \in [d]$ can be written as

$$v_k^{[t+1]} \leftarrow v_k^{[t]} - \eta s_t x_k^{(i_t)} v_k^{[t]} - \eta^{[t]}\left((v^{[t]\odot 2} - v^{\star\odot 2})^\top x^{(i_t)}\right) x_k^{(i_t)} v_k^{[t]}, \tag{8}$$

where $s_t$ is sampled from $\{-\delta, \delta\}$ and $i_t$ is sampled from $[n]$. Let $g_k^{(i_t)} \triangleq ((v^{[t]\odot 2} - v^{\star\odot 2})^\top x^{(i_t)})x_k^{(i_t)}$ be the component coming from the stochastic gradient. Using the fact that $\phi(ab) = \phi(a)\phi(b)$ for any $a, b > 0$, we can evaluate the potential function at time $t+1$,

$$\mathbb{E}\left[\phi(v_k^{[t+1]})\right] = \mathbb{E}\left[\phi(v_k^{[t]})\phi(1 - \eta s_t x_k^{(i_t)} - \eta g_k^{(i_t)})\right] = \phi(v_k^{[t]})\mathbb{E}\left[\phi(1 - \eta s_t x_k^{(i_t)} - \eta g_k^{(i_t)})\right]. \tag{9}$$

Here the expectation is over $s_t$ and $i_t$. We perform Taylor-expansion on the term $\phi(1 - \eta s_t x_k^{(i_t)} - \eta g_k^{(i_t)})$ to deal with the non-linearity and use the fact that $\eta s_t x_k^{(i_t)}$ is mean-zero:

$$\mathbb{E}\left[\phi(1 - \eta s_t x_k^{(i_t)} - \eta g_k^{i_t})\right] \approx \phi(1) - \phi'(1)\eta\mathbb{E}\left[g_k^{i_t}\right] + \frac{1}{2}\phi''(1)\mathbb{E}\left[\left(\eta s_t x_k^{(i_t)} - \eta g_k^{(i_t)}\right)^2\right]$$

$$\leq \phi(1) - \phi'(1)\eta\mathbb{E}\left[g_k^{i_t}\right] + \frac{1}{2}\phi''(1)\mathbb{E}\left[\left(\eta s_t x_k^{(i_t)}\right)^2\right]$$

$$\leq \phi(1) - \phi'(1)\eta\mathbb{E}\left[g_k^{i_t}\right] - \Omega(\eta^2\delta^2). \tag{10}$$

---

[7]In general, any mean-zero noise has a second-order effect on any potential function. Therefore, when the noise level is fixed, as $\eta \to 0$, the effect of the noise diminishes. This is why a lower bound on the learning rate is necessary for the noise to play a role.

In the second line we used $\phi''(1) < 0$ from the concavity and $\mathbb{E}[\eta s_t x_k^{(i_t)}] = 0$, and the third line uses the fact that $s_t \sim \{\pm\delta\}$ and $\mathbb{E}[x_k^{(i_t)2}] \approx 1$ (by the data assumption). The rest of the proof consists of bounding the second term in equation (10) from above to show the potential function is contracting.

We first note for every $i_t$, it holds that $|g_k^{i_t}| \leq \|v^{[t]\odot 2} - v^{\star\odot 2}\|_1 \|x^{(i_t)}\|_\infty^2 \leq (\|v^{[t]}\|_2^2 + r)\|x^{(i_t)}\|_\infty^2$. Furthermore, we can bound the $\ell_2$ norm of $v^{[t]}$ with the following lemma:

**Lemma 3.2.** *In the setting of Theorem 3.1, for some failure probability $\rho > 0$, let $b_0 \triangleq 6\tau d/\rho$. Then, with probability at least $1 - \rho/3$, we have that $\|v^{[t]}\|_2 \leq b_0$ for any $t \leq T_0$.*

Note that $v^{[0]}$ has $\ell_2$ norm $\tau\sqrt{d}$, and here we prove that the norm does not exceed $\tau d$ with high probability. At the first glance, the lemma appears to be mostly auxiliary, but we note that it distinguishes label noise from Gaussian noise, which empirically causes the parameter to blow up as shown in Figure 1. The formal proof is deferred to Section B.

By Lemma 3.2 and the bound on $|g_k^{i_t}|$ in terms of $\|v^{[t]}\|_2$, we have $|g_k^{i_t}| \leq (b_0^2 + r)\|x^{(i_t)}\|_\infty^2 \leq \widetilde{O}(b_0^2 + r)$ with $b_0$ defined in Lemma 3.2 (up to logarithmic factors). Here we use again that each entry of the data is from $\mathcal{N}(0, 1)$. Plugging these into equation (10) we obtain

$$\mathbb{E}\left[\phi(1 - \eta s_t x_k^{(i_t)} - \eta g_k^{i_t})\right] \leq 1 + \eta\widetilde{O}(b_0^2 + r) - \Omega(\eta^2\delta^2) < 1 - \Omega(\eta^2\delta^2)$$

where in the last inequality we use the lower bound on $\eta$ to conclude $\eta^2\delta^2 \gtrsim \eta\widetilde{O}(b_0^2 + r)$. Therefore, summing equation (9) over all the dimensions shows that the potential function decreases exponentially fast: $\mathbb{E}[\Phi(v^{[t+1]})] < (1 - \Omega(\eta^2\delta^2))\Phi(v^{[t]})$. After $T \approx \log(d)/(\eta^2\delta^2)$ iterations, $v^{[T]}$ will already converge to a position such that $\mathbb{E}[\Phi(v^{[T]})] \lesssim \sqrt{1/d}$, which implies $\|v^{[T]}\|_\infty \lesssim 1/d$ with probability at least $1 - \rho$ and finishes the proof.

### 3.3 STAGE 1: GETTING CLOSER TO $v^\star$ WITH ANNEALED LEARNING RATE

Theorem 3.1 shows that the noise decreases the $\infty$-norm of $v$ to $1/d$. This means that $\ell_1$ or $\ell_2$-norm of $v$ is similar to or smaller than that of $v^\star$ if $r$ is constant, and we are in a small-norm region where overfitting is less likely to happen. In the next stage, we anneal the learning rate to slightly reduce the bias of the label noise and increase the contribution of the signal. Recall that $v^\star$ is a sparse vector with support $S \subset [d]$. The following theorem shows that, after annealing the learning rate (from the order of $1/\delta^2$ to $1/\delta$), SGD with label noise increases entries in $v_S$ and decreases entries in $v_{\bar{S}}$ simultaneously, provided that the initialization has $\ell_\infty$-norm bounded by $1/d$. (For simplicity and self-containedness of the statement, we reset the time step to 0.)

**Theorem 3.3.** *In the setting of Section 2.1, given a target error bound $\epsilon_1 > 0$, we assume that $n \geq \widetilde{\Theta}(r^2 \log^2(1/\epsilon_1))$. We run SGD with label noise (Algorithm 1) with an initialization $v^{[0]}$ whose entries are all in $[\epsilon_{\min}, 1/d]$, where $\epsilon_{\min} \geq \exp(-\widetilde{O}(1))$. Let noise level $\delta \geq \widetilde{\Theta}(\log(1/\epsilon_1))$ and learning rate $\eta = \widetilde{\Theta}(1/\delta^2)$, and number of iterations $T = \widetilde{\Theta}(\log(1/\epsilon_1)/\eta)$. Then, with probability at least $0.99$, after $T$ iterations, we have*

$$\|v_S^{[T]} - v_S^\star\|_\infty \leq 0.1 \text{ and } \|v_{\bar{S}}^{[T]} - v_{\bar{S}}^\star\|_1 \leq \epsilon_1. \tag{11}$$

We remark that even though the initialization is relatively small in this stage, the label noise still helps alleviate the reliance on small initialization. Li et al. (2017); Vaskevicius et al. (2019) showed that GD converges to the ground truth with sufficiently small initialization, which is required to be smaller than target error $\epsilon_1$. In contrast, our result shows that with label noise, the initialization does not need to depend on the target error, but only need to have an $\ell_\infty$-norm bound on the order of $1/d$. In other words, $v$ gets closer to $v^\star$ on both $S$ and $\bar{S}$ in our case, whereas in Li et al. (2017); Vaskevicius et al. (2019) the $v_{\bar{S}}$ grows slowly.

The proof of this theorem balances the contribution of the gradient against that of the noise on $S$ and $\bar{S}$. On $S$, the gradient provides a stronger signal than label noise, whereas on $\bar{S}$, the implicit bias of the noise, similar to the effect in Section 3.2, outweighs the gradient and reduces the entries to zero. The analysis is more involved than that of Theorem 3.1, and we defer the full proof to Section C.

**Stage 2: Convergence to the ground-truth $v^\star$:** The conclusion of Theorem 3.3 still allows constant error in the support, namely, $\|v_S - v_S^\star\|_\infty \leq 1/10$. In Theorem D.1, we show that further annealing the learning rate will let the algorithm fully converge to $v^\star$ with any target error $\epsilon$.

*Proof of Theorem 2.1.* In Section E of Appendix, we combine Theorem 3.1, Theorem 3.3, and Theorem D.1 to prove our main Theorem 2.1.

## 4  CONCLUSION

In this work, we study the implicit bias effect induced by noise. For a quadratically-parameterized model, we theoretically show that the parameter-dependent noise has a strong implicit bias, which can help recover the sparse ground-truth from limited data. In comparison, our negative result shows that such a bias cannot be induced by spherical Gaussian noise. Our result explains the empirical observation that replacing mini-batch noise or label noise with Gaussian noise usually leads to degradation in the generalization performance of deep models.

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

# A    EXPERIMENTAL DETAILS

## A.1    EXPERIMENTAL DETAILS FOR THE QUADRATICALLY-PARAMETERIZED MODEL

In the experiment of our quadratically-parameterized model, we use a 100-dimensional model with $n = 40$ data randomly sampled from $\mathcal{N}(0, \mathcal{I}_{100 \times 100})$. We set the first 5 dimensions of the ground-truth $v^\star$ as 1, and the rest dimensions as 0. We always initialize with $v^{[0]} = \mathbb{1}$. We use a constant learning rate $0.01$ for all the experiments except for label noise. For label noise, we start from $0.01$ and then decay the learning rate by a factor of 10 after $1 \times 10^5$ and $2 \times 10^5$ iterations. For "full batch" experiment, we run full batch gradient descent without noise. For "small batch" experiment, in order to fully disentangle the effect of learning rate and mini-batch sgd noise (i.e., to avoid implicit biases from large lr rather than noise), we add small batch noise to full gradient via the following sampling method: for each iteration, we randomly sample two data $i$ and $j$ from $[n]$, and add $\delta(\nabla \ell^{(i)}(v) - \nabla \ell^{(j)}(v))$ to the full gradient (we set $\delta = 1.0$ in our experiment). For label noise, we randomly sample $i \in [n]$ and $s \in \{\delta, -\delta\}$ (we set $\delta = 1.0$ in our experiment), and add noise $\nabla \tilde{\ell}^{(i)}(v) - \nabla \ell^{(i)}(v)$ to full gradient, where $\tilde{\ell}^{(i)}(v) \triangleq \frac{1}{4}(f_v(x^{(i)}) - y^{(i)} - s)^2$. For Gaussian noise experiments, we add noise $\xi \sim \mathcal{N}(0, \sigma^2 \mathcal{I}_{d \times d})$ to full gradient every iteration, where the values of $\sigma$ are shown in Figure 1. For experiments except for Gaussian noises, we train a total of $3 \times 10^5$ iterations. For a more generous comparison, we run all the Gaussian noise experiments for 4 times longer (i.e., $1.2 \times 10^6$ iterations) while plotting them in the same figure after scaling the x-axis by a factor of 4. The test error is measured by the square of $\ell_2$ distance between $v^{\odot 2}$ and $v^{\star \odot 2}$, which is the same as the expectation of loss on a freshly randomly sampled data. The trianing and test error are plotted in Figure 1.

## A.2    EXPERIMENTAL DETAILS FOR DEEP NEURAL NETWORKS ON CIFAR100

We train a VGG19 model (Simonyan and Zisserman, 2014) on CIFAR100, using a small and large batch baseline. We also experiment with adding Gaussian noise to the parameters after every gradient update as well as adding label noise in the following manner: with some probability that depends on the current iteration count, we replace the original label with a randomly chosen one.

To add additional mean-zero noise to the gradient which simulates the effect of label noise in the regression setting, we compute a noisy gradient of the cross-entropy loss $\ell_{ce}$ with respect to model output $f(x)$ as follows:

$$\tilde{\nabla}_{f(x)} \ell_{ce}(f(x), y) = \nabla_{f(x)} \ell_{ce}(f(x), y) + \sigma_{ln} z \tag{12}$$

where $z$ is a 100-dimensional vector (corresponding to each class) distributed according to $\mathcal{N}(0, \mathcal{I}_{100 \times 100})$, and $y$ is the (possibly flipped) label. We backpropagate using this noisy gradient when we compute the gradient of loss w.r.t. parameters for the updates. After tuning, we choose the initial label-flipping probability as $0.1$, and reduce it by a factor of $0.5$ every time the learning rate is annealed. We choose $\sigma_{ln}$ such that $\sigma_{ln} \sqrt{\mathbb{E}[\|z\|_2^2]} = 0.1$, and also decrease $\sigma_{ln}$ by a factor of $0.5$ every time the learning rate is annealed.

To add spherical Gaussian noise to the parameter every update, we simply set $W \leftarrow W + \sigma z$ after every gradient update, where $z$ is a mean-zero Gaussian whose coordinates are drawn independently from $\mathcal{N}(0, 1)$. We tune this $\sigma$ over the values shown in Figure 1.

We turn off weight decay and BatchNorm to isolate the regularization effects of just the noise alone. Standard data augmentation is still present in our runs. Our small batch baseline uses a batch size of 26, and our large batch baseline uses a batch size of 256. In runs where we add noise, the batch size is always 256. For all runs, we use an initial learning rate of 0.004. We train for 410550 iterations (i.e., minibatches), annealing the learning rate by a factor of 0.1 at the 175950-th and 293250-th iteration. Our models take around 20 hours to train on a single NVIDIA TitanXp GPU when the batch size is 256. The final performance gap between label noise or small minibatch training v.s. large batch or Gaussian noise is around 13% accuracy.

### A.3 ADDITIONAL PLOTS

Here we show empirical evidence that training with Gaussian noise fails to converge to a stationary distribution. We train VGG19 network on CIFAR100, and plot the norm of model weight along the training trajectory. As shown in Figure 2, the weigth norm of large batch (LB) and large batch with label noise (LB+LN) both converge to some finite value, while the weight norm of large batch with Gaussian noise (LB+GN) keeps increasing and fails to converge.

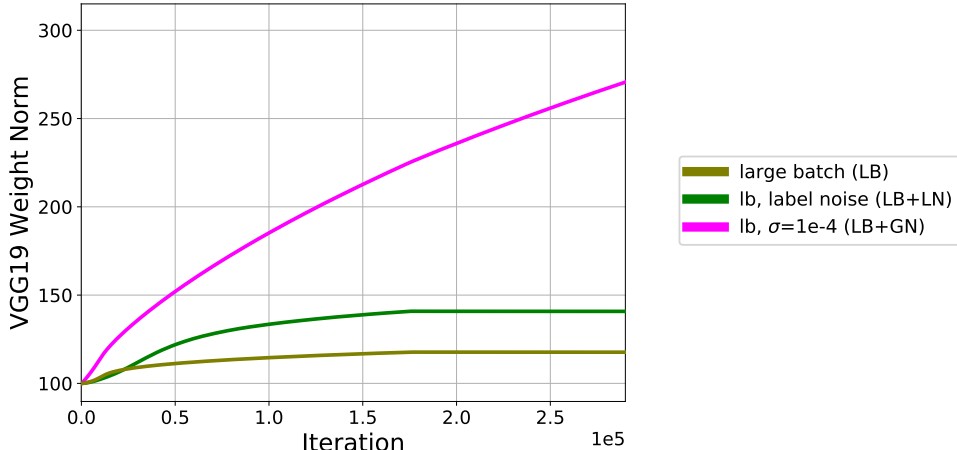

Figure 2: **The norm of model weight along training trajectory.** We demonstrate that the model weight fails to converge when training with Gaussian noise. In contrast, the weigth norm converges for training with label noise or without noise.

## B  PROOF OF THEOREM 3.1

In this section, we will first prove several lemmas on which the proof of Theorem 3.1 is built upon. Then we will provide a proof of Theorem 3.1.

**Definition B.1.** *(b-bounded coupling) Let $v^{[0]}, v^{[1]}, \cdots, v^{[T]}$ be a trajectory of label noise gradient descent with initialization $v^{[0]}$. We call the following random sequence $\tilde{v}^{[t]}$ a b-bounded coupling of $v^{[t]}$: starting from $\tilde{v}^{[0]} = v^{[0]}$, for each time $t < T$, if $\left\|\tilde{v}^{[t]}\right\|_1 \leq b$, we let $\tilde{v}^{[t+1]} \triangleq v^{[t+1]}$; otherwise if $\left\|\tilde{v}^{[t]}\right\|_1 > b$ we don't update, i.e., $\tilde{v}^{[t+1]} \triangleq \tilde{v}^{[t]}$.*

**Lemma B.2.** *In the setting of Theorem 3.1, assume $\left\|x^{(i)}\right\|_\infty \leq b_x$ for any $i \in [n]$. Let $\eta \leq \frac{\rho}{6Tb_x^2(b_0^2+r)}$, where $b_0 = \frac{6\tau d}{\rho}$. Let $\tilde{v}^{[t]}$ be the $b_0$-bounded coupling of $v^{[t]}$. If $\tilde{v}^{[t]}$ is always positive on each dimension, then with probability at least $1 - \frac{\rho}{3}$, there is*

$$\left\|\tilde{v}^{[T]}\right\|_1 \leq b_0. \tag{13}$$

*Proof of Lemma B.2.* Recall the update at $t$-th iteration is:

$$v^{[t+1]} = v^{[t]} - \eta((v^{[t]\odot 2} - v^{\star\odot 2})^\top x^{(i_t)})x^{(i_t)} \odot v^{[t]} - \eta s_t x^{(i_t)} \odot v^{[t]}. \tag{14}$$

We first bound the increase of $\left\|\tilde{v}^{[t]}\right\|_1$ in expectation. When $\left\|\tilde{v}^{[t]}\right\|_1 \leq b_0$, there is:

$$\mathbb{E}\left[\tilde{v}_k^{[t+1]}\right] = \tilde{v}_k^{[t]} - \eta\mathbb{E}[((\tilde{v}^{[t]\odot 2} - v^{\star\odot 2})^\top x^{(i)})x_k^i \tilde{v}_k^{[t]}] \tag{15}$$

$$\leq \tilde{v}_k^{[t]} + \eta(\left\|\tilde{v}^{[t]\odot 2}\right\|_1 + \left\|v^{\star\odot 2}\right\|_1)b_x^2 \tilde{v}_k^{[t]} \tag{16}$$

$$\leq \tilde{v}_k^{[t]} + \eta(b_0^2 + r)b_x^2 \tilde{v}_k^{[t]} \tag{17}$$

where the first inequality is because we can separate the last term into $v^{[t]\odot 2}$ part and $v^{\star\odot 2}$ part and bound them with $\left\|v^{[t]\odot 2}\right\|_1$ and $\left\|v^{\star\odot 2}\right\|_1$ respectively, the second inequality is by $\left\|v^{[t]}\right\|_2^2 \leq \left\|v^{[t]}\right\|_1^2$ and sparsity of $v^\star$. So summing over all dimensions we have $\mathbb{E}\left[\left\|\tilde{v}^{[t+1]}\right\|_1\right] \leq \left\|\tilde{v}^{[t]}\right\|_1 + \eta b_0 b_x(b_0^2 + r)$. This bound is obviously also true when $\left\|\tilde{v}^{[t]}\right\|_1 > b_0$, in which case $\tilde{v}^{[t+1]} = \tilde{v}^{[t]}$.

We then bound the probability of $\left\|\tilde{v}^{[T]}\right\|_1$ being too large:

$$\Pr\left(\left\|\tilde{v}^{[T]}\right\|_1 > b_0\right) \leq \frac{\mathbb{E}\left[\left\|\tilde{v}^{[T]}\right\|_1\right]}{b_0} \tag{18}$$

$$\leq \frac{\tau d + T\eta b_0 b_x^2(b_0^2 + r)}{b_0} \tag{19}$$

$$\leq \frac{\rho}{3}, \tag{20}$$

where the first inequality is Markov Inequality, the second is by the previous equation, and the third is by assumption of $\eta$ and the definition of $b_0$. $\qquad\square$

*Proof of Lemma 3.2.* Notice that when $\left\|\tilde{v}^{[T]}\right\|_1 \leq b_0$, there is $v^{[T]} = \tilde{v}^{[T]}$, Lemma 3.2 naturally follows from Lemma B.2. $\qquad\square$

**Definition B.3.** *(b-bounded potential function) For a vector $v$ that is positive on each dimension, we define the b-bounded potential function $\Phi(v)$ as follows: if $\|v\|_1 \leq b$, we let $\Phi(v) \triangleq \sum_{k=1}^d \sqrt{v_k}$; otherwise $\Phi(v) \triangleq 0$.*

**Lemma B.4.** *In the setting of Theorem 3.1, let $\epsilon_0 = 1/d$. Assume $\left\|x^{(i)}\right\|_\infty \leq b_x$ for $i \in [n]$ with some $b_x > 0$, $\mathbb{E}_i[(x_k^{(i)})^2] \geq \frac{2}{3}$ for all $k \in [d]$. Let $b_0 = \frac{6\tau d}{\rho}$. Assume $\eta\delta b_x + \eta(b_0^2 + r)b_x^2 \leq \frac{1}{16}$, $\eta\delta^2 \geq 32(b_0^2 + r)b_x^2$ and $T = \lceil \frac{32}{\eta^2\delta^2} \log(\frac{\rho\sqrt{\epsilon_0}}{3d\sqrt{\tau}}) \rceil$. Let $\tilde{v}^{[t]}$ be the $b_0$-bounded coupling of $v^{[t]}$, and $\Phi(\cdot)$ is the $b_0$-bounded potential function. If $\tilde{v}^{[t]}$ is always positive on each dimension, then with probability at least $1 - \frac{\rho}{3}$, there is*

$$\Phi(\tilde{v}^{[T]}) \leq \sqrt{\epsilon_0}. \tag{21}$$

*Proof of Lemma B.4.* We first show $\Phi(\tilde{v}^{[t]})$ decreases exponentially in expectation. If $\left\|\tilde{v}^{[t]}\right\|_1 \leq b_0$, we have:

$$\mathbb{E}\left[\Phi(\tilde{v}^{[t+1]})\right] \leq \sum_{k=1}^{d} \mathbb{E}\left[\sqrt{\tilde{v}_k^{[t+1]}}\right] \tag{22}$$

$$= \sum_{k=1}^{d} \mathbb{E}_{s_t, i_t}\left[\sqrt{\tilde{v}_k^{[t]} - \eta s_t x_k^{(i_t)} \tilde{v}_k^{[t]} - \eta((\tilde{v}^{[t]\odot 2} - v^{\star \odot 2})^\top x^{(i_t)}) x_k^{(i_t)} \tilde{v}_k^{[t]}}\right] \tag{23}$$

$$\leq \sum_{k=1}^{d} \sqrt{\tilde{v}_k^{[t]}} \mathbb{E}_{s_t, i_t}\left[\sqrt{1 + \eta s_t x_k^{(i_t)} + \eta(b_0^2 + r) b_x^2}\right] \tag{24}$$

$$\tag{25}$$

where the second inequality is because $\left\|\tilde{v}^{[t]}\right\|_2^2 = \left\|\tilde{v}^{[t]}\right\|_1^2 \leq b_0^2$. Toward bounding the expectation, we notice that by Taylor expansion theorem, there is for any general function $g(x) = \sqrt{1+x}$, there is

$$g(1+x) \leq g(1) + g'(1)x + \frac{1}{2}g''(1)x^2 + \frac{M}{6}|x|^3, \tag{26}$$

where $M$ is upper bound on $|g'''(1+x')|$ for $x'$ in 0 to $x$, which is less than 3 if $|x| \leq \frac{1}{2}$. So in our theorem if $\Delta \triangleq \eta s_t x_k^{(i_t)} + \eta(b_0^2 + r) b_x^2 \in [-\frac{1}{2}, \frac{1}{2}]$, we have

$$\sqrt{1 + \eta s_t x_k^{(i_t)} + \eta(b_0^2 + r) b_x^2} \leq 1 + \frac{1}{2}\Delta - \frac{1}{8}\Delta^2 + \frac{1}{2}|\Delta|^3. \tag{27}$$

Also since $\mathbb{E}_{s_t, i_t}[\Delta] = \eta(b_0^2 + r) b_x^2$, $\mathbb{E}_{s_t, i_t}[\Delta^2] \geq \eta^2 \delta^2 \mathbb{E}_{i_t}[(x_k^{(i_t)})^2] \geq \frac{2}{3}\eta^2 \delta^2$, we have when $|\Delta| \leq \frac{1}{16}$ and $\eta \delta^2 \geq 32(b_0^2 + r) b_x^2$, we have $\mathbb{E}_{s_t, i_t}[\sqrt{1 + \Delta}] \leq 1 - \mathbb{E}_{s_t, i_t}[1 - \frac{1}{16}\Delta^2] \leq 1 - \frac{1}{32}\eta^2 \delta^2$. So

$$\mathbb{E}\left[\Phi(\tilde{v}^{[t+1]})\right] \leq (1 - \frac{1}{32}\eta^2 \delta^2)\Phi(\tilde{v}^{[t]}). \tag{28}$$

Also notice that when $\left\|\tilde{v}^{[t]}\right\|_1 > b_0$, there is $\Phi(\tilde{v}^{[t+1]}) = \Phi(\tilde{v}^{[t]}) = 0$, so obviously we have $\mathbb{E}[\Phi(\tilde{v}^{[t+1]})] \leq (1 - \frac{1}{32}\eta^2 \delta^2)\Phi(\tilde{v}^{[t]})$ always true.

Next we prove that $\Phi(\tilde{v}^{[T]}) \leq \sqrt{\epsilon_0}$ with probability more than $1 - \frac{\rho}{2}$. This is because:

$$\Pr\left(\Phi(\tilde{v}^{[T]}) > \sqrt{\epsilon_0}\right) \leq \frac{\mathbb{E}\left[\Phi(\tilde{v}^{[T]})\right]}{\sqrt{\epsilon_0}} \tag{29}$$

$$\leq \frac{(1 - \frac{1}{32}\eta^2 \delta^2)^T d\sqrt{\tau}}{\sqrt{\epsilon_0}} \tag{30}$$

$$\leq \frac{\rho}{3}. \tag{31}$$

where the first inequality if by Markov Inequaltiy, the second inequality is by the previous inequality, and the last inequality is because $T = \lceil \frac{32}{\eta^2 \delta^2} \log(\frac{3d\sqrt{\tau}}{\rho\sqrt{\epsilon_0}}) \rceil$. □

*Proof of Theorem 3.1.* Let $\rho = 0.01$, $\epsilon_0 = 1/d$. By Lemma G.1, and Lemma G.2, when $n \geq \Theta(\log d)$, with probability at least $1 - \frac{\rho}{3}$ there is $\left\|x^{(i)}\right\|_\infty \leq b_x$ for all $i \in [n]$ with some $b_x = \Theta(\sqrt{\log(nd)})$, and $\mathbb{E}_i[(x_k^{(i)})^2] \geq \frac{2}{3}$ for all $k \in [d]$.

Let $b_0 = \frac{6\tau d}{\rho}$. We try to define $\eta$ and $\delta$ such that when $T = \lceil \frac{32}{\eta^2 \delta^2} \log(\frac{3d\sqrt{\tau}}{\rho\sqrt{\epsilon_0}}) \rceil$, the assumptions $\eta \leq \frac{\rho}{6Tb_x^2(b_0^2+r)}$ and $\tilde{v}^{[t]}$ always being positive in Lemma B.2 and assumptions $\eta\delta b_x + \eta(b_0^2 + r) b_x^2 \leq \frac{1}{16}$ and $\eta\delta^2 \geq 32(b_0^2 + r) b_x^2$ in Lemma B.4 are satisfied.

Assume $\delta \geq 6 \times 32^2 b_x^3 \frac{(b_0^2 + r)}{\rho} \log(\frac{3d\sqrt{\tau}}{\rho\sqrt{\epsilon_0}})$, then we only need $\eta \in \left[\frac{6 \times 32 b_x^2}{\rho\delta^2}(b_0^2 + r)\log(\frac{3d\sqrt{\tau}}{\rho\sqrt{\epsilon_0}}), \frac{1}{32\delta b_x}\right]$, and then all the above assumptions are satisfied.

Let $\tilde{v}^{[t]}$ be the $b_0$-bounded coupling of $v^{[t]}$. According to Lemma B.4, we know with probability at least $1 - \frac{\rho}{3}$, $\Phi(\tilde{v}^{[T]}) \leq \sqrt{\epsilon_0}$, which means that either $\sum_{k=1}^{d} \sqrt{\tilde{v}_k^{[T]}} \leq \sqrt{\epsilon_0}$ or $\left\|\tilde{v}^{[T]}\right\|_1 > b_0$. According to Lemma B.2, we know with probability at most $\frac{\rho}{3}$, $\left\|\tilde{v}^{[T]}\right\|_1 > b_0$. Combining these two statements, we know with probability at least $1 - \frac{2\rho}{3}$, $\left\|\tilde{v}^{[T]}\right\|_1 \leq b_0$ and $\sum_{k=1}^{d} \sqrt{\tilde{v}_k^{[T]}} \leq \sqrt{\epsilon_0}$. Notice that $\left\|\tilde{v}^{[T]}\right\|_1 \leq b_0$ implies $v^{[T]} = \tilde{v}^{[T]}$, while $\sum_{k=1}^{d} \sqrt{\tilde{v}_k^{[T]}} \leq \sqrt{\epsilon_0}$ implies $\tilde{v}_k^{[T]} \leq \epsilon_0$ for all dimension $k$, so we've finished the proof for the upper bound.

We then give a lower bound for each dimension of $\tilde{v}^{[T]}$. We can bound the decrease of any dimension $k$ at time $t$:

$$\tilde{v}_k^{[t+1]} \geq (1 - \eta\delta - \eta(b_0^2 + r))\tilde{v}_k^{[t]} \tag{32}$$

$$\geq (1 - 2\eta\delta)\tilde{v}_k^{[t]}. \tag{33}$$

where the first inequality is by update rule and the second is because $\delta > (b_0^2 + r)$. Putting in the value of $T$, we have

$$\tilde{v}_k^{[T]} \geq (1 - 2\eta\delta)^T \tau \tag{34}$$

$$> \exp\left(-\frac{64}{\eta\delta}\log(\frac{3d\sqrt{\tau}}{\rho\sqrt{\epsilon_0}})\right). \tag{35}$$

$\square$

## C   PROOF OF THEOREM 3.3

In this section, we will first prove several lemmas on which the proof of Theorem 3.3 is built upon. Then we will provide a proof of Theorem 3.3.

**Definition C.1.** *($(b, \epsilon)$-bounded coupling) Let $v^{[0]}, v^{[1]}, \cdots, v^{[T]}$ be a trajectory of label noise gradient descent with initialization $v^{[0]}$. Recall $S \subset [d]$ is the support set of $v^\star$, we notate $\tilde{v}_S^{[t]}$ a $r$-dimensional vector composed with those dimensions in $S$ of $\tilde{v}^{[t]}$, and $\tilde{v}_{\bar{S}}^{[t]}$ the other $d - r$ dimensions. We call the following random sequence $\tilde{v}^{[t]}$ a $(b, \epsilon)$-bounded coupling of $v^{[t]}$: starting from $\tilde{v}^{[0]} = v^{[0]}$, for each time $t < T$, if $\left\| \tilde{v}_{\bar{S}}^{[t]} \right\|_1 \le \epsilon$ and $\left\| \tilde{v}_S^{[t]} \right\|_\infty \le b$, we let $\tilde{v}^{[t+1]} \triangleq v^{[t+1]}$; otherwise $\tilde{v}^{[t+1]} \triangleq \tilde{v}^{[t]}$.*

**Lemma C.2.** *In the setting of Theorem 3.3, let $\rho \triangleq \frac{1}{100}$, $c_1 \triangleq \frac{1}{10}$, $\tilde{\epsilon}_1 \triangleq \frac{12}{\rho}$, $C_x \triangleq \max_{j \ne k} |\mathbb{E}_i[x_j^{(i)} x_k^{(i)}]|$. Assume $\left\| x^{(i)} \right\|_\infty \le b_x$ for $i \in [n]$ for some $b_x > 0$, and $\mathbb{E}_i[(x_k^{(i)})^2] \ge \frac{2}{3}$ for $k \in [d]$. Let $\tilde{v}^{[t]}$ be a $(1 + c_1, \tilde{\epsilon}_1)$-bounded coupling of $v^{[t]}$. Assume $\frac{c_1^2}{8\eta\delta^2 b_x^2} \ge \log \frac{6rT^2}{\rho}$, $(\tilde{\epsilon}_1^2 + r)C_x b_x^2 \le \frac{c_1}{20}$ and $\delta \ge b_x(\tilde{\epsilon}_1^2 + r)$. Then, with probability at least $1 - \frac{\rho}{6}$, there is $\left\| \tilde{v}_S^{[T]} \right\|_\infty \le 1 + c_1$.*

*Proof of Lemma C.2.* For any fixed $1 \le t_1 < t_2 \le T$ and dimension $k \in S$, we consider the event that $\tilde{v}^{[t_1]} \in [1 + \frac{c_1}{3}, 1 + \frac{c_1}{2}]$, and at time $t_2$ it is the first time in the trajectory such that $\tilde{v}^{[t_2]} > 1 + c_1$. We first bound the probability of this event happens, i.e., the following quantity:

$$\Pr\left( \tilde{v}_k^{[t_2]} > 1 + c_1 \wedge \tilde{v}_k^{[t_1]} \le 1 + \frac{c_1}{2} \wedge \tilde{v}_k^{[t_1:t_2]} \in [1 + \frac{c_1}{3}, 1 + c_1] \right), \tag{36}$$

where $\tilde{v}_k^{[t_1:t_2]} \in [1 + \frac{c_1}{3}, 1 + c_1]$ means that for all $t$ such that $t_1 \le t < t_2$, there is $1 + \frac{c_1}{3} \le \tilde{v}_k^{[t]} \le 1 + c_1$.

Notice that when $\left\| \tilde{v}_{\bar{S}}^{[t]} \right\|_1 \le \tilde{\epsilon}_1$ and $\left\| \tilde{v}_S^{[t]} \right\|_\infty \le 1 + c_1$ and $\tilde{v}_k^{[t_1:t+1]} \in [1 + \frac{c_1}{3}, 1 + c_1]$, there is

$$\mathbb{E}[\tilde{v}_k^{[t+1]} - 1] = \mathbb{E}_{s_t, i_t} \left[ \left( 1 + \eta s_t x_k^{i_t} - \eta((\tilde{v}^{[t] \odot 2} - v^{\star \odot 2})^\top x^{(i_t)}) x_k^{(i_t)} \right) \tilde{v}_k^{[t]} - 1 \right] \tag{37}$$

$$\le (\tilde{v}_k^{[t]} - 1) - \frac{2}{3} \eta \tilde{v}_k^{[t]} (\tilde{v}_k^{[t]} + 1)(\tilde{v}_k^{[t]} - 1) + \eta(\tilde{\epsilon}_1^2 C_x + r C_x) b_x^2 \tilde{v}_k^{[t]} \tag{38}$$

$$\le (1 - \eta)(\tilde{v}_k^{[t]} - 1). \tag{39}$$

where the first inequality is because $\left\| \tilde{v}_{\bar{S}}^{[t]} \right\|_2^2 \le \tilde{\epsilon}_1^2$ and $\mathbb{E}_{i_t}[(x_k^{(i)})^2] \ge \frac{1}{2}$, the second inequality is because $(\tilde{\epsilon}_1^2 + r) b_x^2 C_x \le \frac{c_1}{20}$. Also, we can bound the variance of this martingale as

$$\text{Var}\left[ \tilde{v}_k^{[t+1]} - 1 \mid \tilde{v}^{[t]} - 1 \right] = \text{Var}\left[ \eta s_t x_k^{(i_t)} \tilde{v}_k^{[t]} \right] + \text{Var}\left[ \eta((\tilde{v}^{[t] \odot 2} - v^{\star \odot 2})^\top x^{(i_t)}) x_k^{(i_t)} \tilde{v}_k^{[t]} \right] \tag{40}$$

$$\le (\eta \delta b_x (1 + c_1))^2 + \eta^2 (\tilde{\epsilon}_1^2 + r)^2 b_x^4 (1 + c_1)^2 \tag{41}$$

$$\le 4\eta^2 \delta^2 b_x^2, \tag{42}$$

where the first inequality is because $\eta s_t x_k^{(i_t)} \tilde{v}_k^{[t]}$ is mean-zero, the second inequality is by $\left\| x^{(i)} \right\|_\infty \le b_x$, the third inequality is by $\delta \ge b_x(\tilde{\epsilon}_1^2 + r)$.

By Lemma G.3, we have

$$\Pr(\tilde{v}_k^{[t_2]} - 1 > c_1) \tag{43}$$

$$\le e^{\frac{-c_1^2}{8\eta^2 \delta^2 b_x^2 \sum_{t=0}^{t_2 - t_1 - 1} (1 - \eta)^{2t}}} \tag{44}$$

$$\le e^{\frac{-c_1^2}{8\eta\delta^2 b_x^2}}, \tag{45}$$

where the first inequality is by Lemma G.3, the second inequality is by taking the sum of denominator.

Finally, we finish the proof with a union bound. Since if $\left\|\tilde{v}_S^{[T]}\right\|_\infty > 1 + c_1$, the event in Equation 36 has to happen for some $k \in S$ and $1 \le t_1 < t_2 \le T$, so we have

$$\Pr\left(\left\|\tilde{v}_S^{[T]}\right\|_\infty > 1 + c_1\right) \tag{46}$$

$$\le \sum_{k \in S} \sum_{1 \le t_1 < t_2 \le T} \Pr\left(\tilde{v}_k^{[t_2]} > 1 + c_1 \wedge \tilde{v}_k^{[t_1]} \le 1 + \frac{c_1}{2} \wedge \tilde{v}_k^{[t_1:t_2]} \in [1 + \frac{c_1}{3}, 1 + c_1]\right) \tag{47}$$

$$\le rT^2 e^{\frac{-c_1^2}{8\eta\delta^2 b_x^2}} \tag{48}$$

$$\le \frac{\rho}{6}, \tag{49}$$

where the last inequality is by assumption. $\qquad\square$

**Lemma C.3.** *In the setting of Lemma C.2, assume $(\tilde{\epsilon}_1^2 + r)C_x \le \frac{\rho}{12T\eta b_x^2}$. Then, with probability at least $1 - \frac{\rho}{6}$, there is $\left\|\tilde{v}_{\bar{S}}^{[T]}\right\|_1 \le \tilde{\epsilon}_1$.*

*Proof of Lemma C.3.* We first bound the increase of $\left\|\tilde{v}_{\bar{S}}^{[t]}\right\|_1$ in expectation. When $\left\|\tilde{v}_S^{[t]}\right\|_\infty \le 1 + c_1$ and $\left\|\tilde{v}_{\bar{S}}^{[t]}\right\|_1 \le \tilde{\epsilon}_1$, for any $k \notin S$, there is:

$$\mathbb{E}\left[\tilde{v}_k^{[t+1]}\right] = \tilde{v}_k^{[t]} - \eta\mathbb{E}_i\left[((\tilde{v}^{[t]\odot 2} - v^{\star\odot 2})^\top x^{(i)})x_k^{(i)}\tilde{v}_k^{[t]}\right] \tag{50}$$

$$\le \tilde{v}_k^{[t]} + \eta(\tilde{\epsilon}_1^2 + r)C_x b_x^2 \tilde{v}_k^{[t]}. \tag{51}$$

because we can bound the dimensions in $S$ and those not in $S$ respectively. So summing over all dimensions not in $S$ we have $\mathbb{E}\left[\left\|\tilde{v}_{\bar{S}}^{[t+1]}\right\|_1\right] \le \left\|\tilde{v}_{\bar{S}}^{[t]}\right\|_1 + \eta(\tilde{\epsilon}_1^2 + r)C_x b_x^2 \tilde{\epsilon}_1$. This bound is obviously also true when $\left\|\tilde{v}_S^{[t]}\right\|_\infty > 1 + c_1$ and $\left\|\tilde{v}_{\bar{S}}^{[t]}\right\|_1 > \tilde{\epsilon}_1$.

We then bound the probability of $\left\|\tilde{v}_{\bar{S}}^{[T]}\right\|_1$ being too large:

$$\Pr\left(\left\|\tilde{v}_{\bar{S}}^{[T]}\right\|_1 > \tilde{\epsilon}_1\right) \le \frac{\mathbb{E}\left[\left\|\tilde{v}_{\bar{S}}^{[T]}\right\|_1\right]}{\tilde{\epsilon}_1} \tag{52}$$

$$\le \frac{1 + T\eta\tilde{\epsilon}_1(\tilde{\epsilon}_1^2 + r)C_x b_x^2}{\tilde{\epsilon}_1} \tag{53}$$

$$\le \frac{\rho}{6}. \tag{54}$$

where the first inequality is Markov Inequality, the second is by $\left\|\tilde{v}_{\bar{S}}^{[0]}\right\|_1 \le 1$ since every dimension is less than $1/d$, the third inequality is because $\tilde{\epsilon}_1 = \frac{12}{\rho}$ and $(\tilde{\epsilon}_1^2 + r)C_x \le \frac{\rho}{12T\eta b_x^2}$. $\qquad\square$

**Lemma C.4.** *In the setting of Lemma C.2, assume $(\tilde{\epsilon}_1^2 + r)C_x b_x^2 < \frac{c_1}{12} - \frac{c_1^2}{4}$, $\eta\delta^2 \le \frac{c_1}{8}$, $T\eta \ge \frac{16}{c_1}\log\frac{1}{\epsilon_{min}}$ and $\frac{T}{\delta^2} \ge \frac{2^9}{c_1^2}\log\frac{6r}{\rho}$. Then, for any $k \in S$, with probability at least $1 - \frac{\rho}{6r}$, either $\max_{t \le T}\tilde{v}_k^{[t]} \ge 1 - \frac{c_1}{2}$, or $\left\|\tilde{v}_S^{[T]}\right\|_\infty > 1 + c_1$, or $\left\|\tilde{v}_{\bar{S}}^{[T]}\right\|_1 > \tilde{\epsilon}_1$.*

*Proof of Lemma C.4.* Fix $k \in S$. Let $\hat{v}^{[t]}$ be the following coupling of $\tilde{v}^{[t]}$: starting from $\hat{v}^{[0]} = \tilde{v}^{[0]}$, for each time $t < T$, if $\left\|\tilde{v}_{\bar{S}}^{[t]}\right\|_1 \le \tilde{\epsilon}_1$ and $\left\|\tilde{v}_S^{[t]}\right\|_\infty \le 1 + c_1$ and $\tilde{v}_k^{[t]} \le 1 - \frac{c_1}{2}$, we let $\hat{v}^{[t+1]} \triangleq \tilde{v}^{[t+1]}$; otherwise $\hat{v}^{[t+1]} \triangleq (1 + \frac{c_1}{2}\eta)\hat{v}^{[t]}$. Intuitively, whenever $\tilde{v}^{[t]}$ exceeds the proper range, we only times $\hat{v}^{[t]}$ by $1 + \frac{c_1}{2}\eta$ afterwards, otherwise we let it be the same as $\tilde{v}^{[t]}$.

We first show that $-t \log(1 + \frac{c_1}{2}\eta) + \log \hat{v}_k^{[t]}$ is a supermartingale, i.e., $\mathbb{E}[\log \hat{v}_k^{[t+1]} \mid \hat{v}^{[t]}] \geq \log(1 + \frac{c_1}{2}\eta) + \log \hat{v}_k^{[t]}$. This is obviously true if $\left\| \tilde{v}_{\bar{S}}^{[t]} \right\|_1 > \tilde{\epsilon}_1$ or $\left\| \tilde{v}_S^{[t]} \right\|_\infty > 1 + c_1$ or $\tilde{v}_k^{[t]} > 1 - \frac{c_1}{2}$. Otherwise, there is

$$\mathbb{E}[\log \hat{v}_k^{[t+1]} \mid \hat{v}^{[t]}] = \mathbb{E}[\log \tilde{v}_k^{[t+1]} \mid \tilde{v}^{[t]}] \tag{55}$$

$$= \mathbb{E}_{s_t, i_t} \left[ \log \left( 1 + \eta s_t - \eta (\tilde{v}^{[t] \odot 2} - v^{\star \odot 2})^\top x^{(i_t)} x_k^{(i_t)} \right) \right] + \log \tilde{v}_k^{[t]} \tag{56}$$

$$\geq \mathbb{E}_{s_t} \left[ \log \left( 1 + \eta s_t + \frac{2}{3}\eta(1 - (\tilde{v}_k^{[t]})^2) - \eta(\tilde{\epsilon}_1^2 + r)C_x b_x^2 \right) \right] + \log \tilde{v}_k^{[t]} \tag{57}$$

$$\geq \log(1 + \frac{c_1}{4}\eta) + \log \tilde{v}_k^{[t]}, \tag{58}$$

where the first inequality is by the update rule, the second inequality is because $(\tilde{\epsilon}_1^2 + r)C_x b_x^2 < \frac{c_1}{12} - \frac{c_1^2}{4}$ and $4\eta\delta^2 \leq \frac{c_1}{2}$ and $\delta \geq \tilde{\epsilon}_1^2 + r$. So by Azuma inequality, we have

$$\Pr\left( \hat{v}_k^{[T]} < 1 - \frac{c_1}{2} \right) \tag{59}$$

$$\leq e^{-\frac{2\left( T \log(1 + \frac{c_1}{4}\eta) + \log \epsilon_{min} - \log(1 - \frac{c_1}{2}) \right)^2}{T(2\eta\delta)^2}} \tag{60}$$

$$\leq e^{-\frac{(\frac{1}{2} T \log(1 + \frac{c_1}{4}\eta))^2}{2T\eta^2\delta^2}} \tag{61}$$

$$\leq e^{-\frac{T c_1^2}{2^9 \delta^2}} \tag{62}$$

$$\leq \frac{\rho}{6r}. \tag{63}$$

where the first inequality is because Azuma inequality and $\text{Var}[\log \hat{v}_k^{[t+1]} \mid \hat{v}^{[t]}] \leq (2\eta\delta)^2$, and the second inequality is because $T \log(1 + \frac{c_1}{4}\eta) \geq 2 \log \frac{1}{\epsilon_{min}}$ which is true because $T\eta \geq \frac{16}{c_1} \log \frac{1}{\epsilon_{min}}$, the third inequality is because $\log(1 + \frac{c_1}{4}\eta) \geq \frac{c_1}{8}\eta$, the last inequality is because $\frac{T}{\delta^2} \geq \frac{2^9}{c_1^2} \log \frac{6r}{\rho}$. $\square$

**Lemma C.5.** *In the setting of Lemma C.2, assume $(\tilde{\epsilon}_1^2 + r)C_x b_x^2 \leq \frac{c_1}{20}$ and $\frac{c_1^2}{8\eta\delta^2} \geq \log \frac{6rT^2}{\rho}$. Then, for any $k \in S$, with probability at least $1 - \frac{\rho}{6r}$, either $\max_{t < T} \tilde{v}_k^{[t]} < 1 - \frac{c_1}{2}$ or $\tilde{v}_k^{[T]} \geq 1 - c_1$.*

*Proof of Lemma C.5.* For any fixed $1 \leq t_1 < t_2 \leq T$ and dimension $k \in S$, we consider the event that $\tilde{v}^{[t_1]} \in [1 - \frac{c_1}{2}, 1 - \frac{c_1}{3}]$, and at time $t_2$ it is the first time in the trajectory such that $\tilde{v}^{[t_2]} > 1 < c_1$. We first bound the probability of this event happens, i.e., the following quantity:

$$\Pr\left( \tilde{v}_k^{[t_2]} < 1 - c_1 \wedge \tilde{v}_k^{[t_1]} \geq 1 - \frac{c_1}{2} \wedge \tilde{v}_k^{[t_1:t_2]} \in [1 - c_1, 1 - \frac{c_1}{3}] \right), \tag{64}$$

where $\tilde{v}_k^{[t_1:t_2]} \in [1 - c_1, 1 - \frac{c_1}{3}]$ means that for all $t$ such that $t_1 \leq t < t_2$, there is $1 - c_1 \leq \tilde{v}_k^{[t]} \leq 1 - \frac{c_1}{3}$.

Notice that when $\left\| \tilde{v}_{\bar{S}}^{[t]} \right\|_1 \leq \tilde{\epsilon}_1$ and $\left\| \tilde{v}_S^{[t]} \right\|_\infty \leq 1 + c_1$ and $\tilde{v}_k^{[t_1:t+1]} \in [1 - c_1, 1 - \frac{c_1}{3}]$,

$$\mathbb{E}[1 - \tilde{v}_k^{[t+1]}] = \mathbb{E}_{s_t, i_t} \left[ 1 - (1 + \eta s_t x_k^{i_t} - \eta(\tilde{v}^{[t] \odot 2} - v^{\star \odot 2})^\top x^{(i_t)} x_k^{(i_t)}) \tilde{v}_k^{[t]} \right] \tag{65}$$

$$\leq (1 - \tilde{v}_k^{[t]}) - \frac{2}{3}\eta \tilde{v}_k^{[t]}(\tilde{v}_k^{[t]} + 1)(1 - \tilde{v}_k^{[t]}) + \eta(\tilde{\epsilon}_1^2 + r)C_x b_x^2 \tilde{v}_k^{[t]} \tag{66}$$

$$\leq (1 - \eta)(1 - \tilde{v}_k^{[t]}). \tag{67}$$

where the first inequality is because $\left\| \tilde{v}_{\bar{S}}^{[t]} \right\|_2^2 \leq \tilde{\epsilon}_1^2$, the second inequality is because $(\tilde{\epsilon}_1^2 + r)C_x b_x^2 \leq \frac{c_1}{20}$. Also, we can bound the variance of this martingale as

$$\text{Var}\left[ 1 - \tilde{v}_k^{[t]} \mid \tilde{v}_k^{[t]} \right] \leq (2\eta\delta)^2. \tag{68}$$

By Lemma G.3, we have

$$\Pr(1 - \tilde{v}_k^{[t_2]} > c_1) \tag{69}$$

$$\leq e^{\frac{-c_1^2}{8\eta^2\delta^2 \sum_{t=0}^{t_2-t_1-1}(1-\eta)^{2t}}} \tag{70}$$

$$\leq e^{\frac{-c_1^2}{8\eta\delta^2}}, \tag{71}$$

where the first inequality is by Lemma G.3, the second inequality is by taking the sum of denominator.

Finally, we finish the proof with a union bound. Since if $\max_{t<T} \tilde{v}_k^{[t]} > 1 - \frac{c_1}{2}$ but $\tilde{v}_k^{[T]} < 1 - c_1$, the event in Equation 64 has to happen for some $1 \leq t_1 < t_2 \leq T$, so we have

$$\Pr\left(\max_{t<T} \tilde{v}_k^{[t]} > 1 - \frac{c_1}{2} \wedge \tilde{v}_k^{[T]} < 1 - c_1\right) \tag{72}$$

$$\leq \sum_{1 \leq t_1 < t_2 \leq T} \Pr\left(\tilde{v}_k^{[t_2]} < 1 - c_1 \wedge \tilde{v}_k^{[t_1]} \geq 1 - \frac{c_1}{2} \wedge \tilde{v}_k^{[t_1:t_2]} \in [1 - c_1, 1 - \frac{c_1}{3}]\right) \tag{73}$$

$$\leq T^2 e^{\frac{-c_1^2}{8\eta\delta^2}} \tag{74}$$

$$\leq \frac{\rho}{6r}. \tag{75}$$

$\square$

**Definition C.6.** *($(b, \epsilon)$-bounded potential function) For a vector $v$ that is positive on each dimension, we define the $(b, \epsilon)$-bounded potential function $\Phi(v)$ as follows: if $\|v_{\bar{S}}\|_1 \leq \epsilon$ and $\|v_S\|_\infty \leq b$, we let $\Phi(v) \triangleq \sum_{k \notin S} \sqrt{v_k}$; otherwise $\Phi(v) \triangleq 0$.*

**Lemma C.7.** *In the setting of Lemma C.2, assume $\frac{2}{3}\eta\delta^2 > 32(\tilde{\epsilon}_1^2 + r)C_x b_x^2$ and $T\eta^2\delta^2 \geq 16\log\left(\frac{6\sqrt{d}}{\rho\sqrt{\epsilon_1}}\right)$. Then, with probability at least $1 - \frac{\rho}{6}$, there is $\Phi(\tilde{v}^{[T]}) \leq \sqrt{\epsilon_1}$.*

*Proof of Lemma C.7.* We first show $\Phi(\tilde{v}^{[t]})$ decreases exponentially in expectation. For any $0 \leq t \leq T$, if $\left\|\tilde{v}_S^{[t]}\right\|_\infty \leq 1 + c_1$ and $\left\|\tilde{v}_{\bar{S}}^{[t]}\right\|_1] \leq \tilde{\epsilon}_1$, we have:

$$\mathbb{E}\left[\Phi(\tilde{v}^{[t+1]})\right] \leq \sum_{k \notin S} \mathbb{E}\left[\sqrt{\tilde{v}_k^{[t+1]}}\right] \tag{76}$$

$$= \sum_{k \notin S} \mathbb{E}_{s_t, i_t}\left[\sqrt{\tilde{v}_k^{[t]} + \eta s_t x_k^{i_t}\tilde{v}_k^{[t]} - \eta(\tilde{v}^{[t]\odot 2} - v^{\star\odot 2})^\top x^{(i_t)} x_k^{(i_t)}\tilde{v}_k^{[t]}}\right] \tag{77}$$

$$\leq \sum_{k \notin S} \sqrt{\tilde{v}_k^{[t]}}\,\mathbb{E}_{s_t, i_t}\left[\sqrt{1 + \eta s_t x_k^{(i_t)} + \eta(\tilde{\epsilon}_1^2 + r)C_x b_x^2}\right] \tag{78}$$

$$\leq (1 - \frac{1}{16}\eta^2\delta^2)\Phi(\tilde{v}^{[t]}), \tag{79}$$

where the second inequality is because $\left\|\tilde{v}^{[t]}\right\|_2^2 \leq \tilde{\epsilon}_1^2$, the last inequality is by Taylor expansion and $\frac{2}{3}\eta\delta^2 > 32(\tilde{\epsilon}_1^2 + r)C_x b_x^2$. Also notice that when $\left\|\tilde{v}_S^{[t]}\right\|_\infty > 1 + c_1$ or $\left\|\tilde{v}_{\bar{S}}^{[t]}\right\|_1 > \tilde{\epsilon}_1$, there is $\Phi(\tilde{v}^{[t+1]}) = p(\tilde{v}^{[t]}) = 0$, so obviously we have $\mathbb{E}[\Phi(\tilde{v}^{[t+1]})] \leq (1 - \frac{1}{16}\eta^2\delta^2)\Phi(\tilde{v}^{[t]})$ always true.

Next we bound the probability of $\Phi(\tilde{v}^{[T]}) \leq \sqrt{\epsilon_1}$:

$$\Pr(\Phi(\tilde{v}^{[T]}) > \sqrt{\epsilon_1}) \leq \frac{\mathbb{E}[\Phi(\tilde{v}^{[T]})]}{\sqrt{\epsilon_1}} \tag{80}$$

$$\leq \frac{(1 - \frac{1}{16}\eta^2\delta^2)^T \sqrt{d}}{\sqrt{\epsilon_1}} \tag{81}$$

$$\leq \frac{e^{-\frac{1}{16}T\eta^2\delta^2} \sqrt{d}}{\sqrt{\epsilon_1}} \tag{82}$$

$$\leq \frac{\rho}{6}. \tag{83}$$

where the first inequality if by Markov Inequaltiy, the second inequality is by the previous inequality and initially $\Phi(v^{[0]}) \leq \sqrt{d}$, the third is by $1 - x \leq e^{-x}$ for any $x \in \mathbb{R}$, and the last inequality is by $T\eta^2\delta^2 \geq 16 \log\left(\frac{6\sqrt{d}}{\rho\sqrt{\epsilon_1}}\right)$.

$\square$

*Proof of Theorem 3.3.* Let $\rho = 0.01$, $c_1 = 0.1$, $\tilde{\epsilon}_1 \triangleq \frac{12}{\rho}$, $C_x \triangleq \max_{j\neq k} |\mathbb{E}_i[x_j^{(i)} x_k^{(i)}]|$. Let $b_x = \sqrt{2\log\frac{30d^2}{\rho}} = \widetilde{\Theta}(1)$. According to Lemma G.1, when $n \leq d$, there is with probability at least $1 - \frac{\rho}{15}$ we have $\|x^{(i)}\|_\infty \leq b_x$ for $i \in [d]$.

Assume $\delta$ be positive number such that $\frac{16}{\delta^2} \log \frac{6\sqrt{d}}{\rho\epsilon_{min}\sqrt{\epsilon_1}} \leq 1$ and $\delta \geq b_x(\tilde{\epsilon}_1^2 + r)$. (since $\epsilon_{min} \geq \exp(-\widetilde{O}(1))$ this means $\delta \geq \widetilde{\Theta}(r + \log(1/\epsilon_1))$ .) Let $P = \frac{c_1^2}{32\delta^2 b_x^2 \log \frac{5r}{\rho}}$, $Q = 2\log\frac{1}{P}$, $\eta = \min\{\frac{P}{Q}, \frac{16}{\delta^2}\} = \widetilde{\Theta}(\frac{1}{\delta^2})$, $T = \frac{16}{\eta^2\delta^2} \log \frac{6\sqrt{d}}{\rho\epsilon_{min}\sqrt{\epsilon_1}} = \widetilde{\Theta}(\log(1/\epsilon_1)/\eta)$. Assume $C_x b_x^2(\tilde{\epsilon}_1^2 + r) \leq \min\left\{\frac{\eta\delta^2}{48}, \frac{\rho}{12T\eta}\right\} = \widetilde{\Theta}(\rho/\log(1/\epsilon_1))$. (this means $C_x \leq \widetilde{\Theta}(\frac{\rho}{r\log(1/\epsilon_1)})$.)

We show the assumptions in the previous lemmas are all satisfied. The assumption $\frac{c_1^2}{8\eta\delta^2 b_x^2} \geq \log\frac{6rT^2}{\rho}$ in Lemma C.2 is satisfied by

$$\frac{c_1^2}{8\eta\delta^2 b_x^2} \geq \log\frac{6rT^2}{\rho} \tag{84}$$

$$\Leftarrow \frac{c_1^2}{8\eta\delta^2 b_x^2} \geq \log\frac{6r}{\rho} + 4\log\frac{1}{\eta} \tag{85}$$

$$\Leftarrow \eta\log\frac{1}{\eta} \leq \frac{c_1^2}{32\delta^2 b_x^2 \log\frac{6r}{\rho}} = P, \tag{86}$$

where the first is by $T \leq \frac{1}{\eta^2}$, the second is by $\log\frac{6r}{\rho} + 4\log\frac{1}{\eta} \leq 4\log\frac{6r}{\rho}\log\frac{1}{\eta}$, and the last line is true because

$$\eta\log\frac{1}{\eta} \leq \frac{P}{Q}\log\frac{Q}{P} \tag{87}$$

$$= P(\frac{\log Q}{Q} + \frac{\log 1/P}{Q}) \tag{88}$$

$$\leq P. \tag{89}$$

The assumption $\delta \geq b_x(\tilde{\epsilon}_1^2 + r)$ in Lemma C.2 is satisfied by definition of $\delta$. The assumption $(\tilde{\epsilon}_1^2 + r)C_x b_x^2 \leq \frac{c_1}{20}$ in Lemma C.2 is satisfied by

$$C_x b_x^2(\tilde{\epsilon}_1^2 + r) \leq \frac{\eta\delta^2}{48} \leq \frac{c_1^2}{96} \leq \frac{c_1}{20}, \tag{90}$$

where we use

$$\eta\delta^2 \leq \delta^2\frac{P}{Q} \leq \frac{c_1^2}{2}. \tag{91}$$

The assumption $(\tilde{\epsilon}_1^2 + r)C_x b_x^2 \leq \frac{\rho}{12T\eta}$ in Lemma C.3 is satisfied by assumption of $C_x$.

The assumption $\frac{T}{\delta^2} \geq \frac{2^9}{c_1^2} \log \frac{6r}{\rho}$ in Lemma C.4 is satisfied by

$$\frac{T}{\delta^2} \geq \frac{16}{\eta^2 \delta^4} \log \frac{6r}{\rho} \geq \frac{2^6}{c_1^4} \log \frac{6r}{\rho} \geq \frac{2^9}{c_1^2} \log \frac{6r}{\rho}. \tag{92}$$

The other two assumptions $(\tilde{\epsilon}_1^2 + r)C_x b_x^2 < \frac{c_1}{12} - \frac{c_1^2}{4}$ and $\eta\delta^2 \leq \frac{c_1}{8}$ in Lemma C.4 follows from $C_x b_x^2 (\tilde{\epsilon}_1^2 + r) < \frac{\eta\delta^2}{48}$ and $\eta\delta^2 \leq \frac{c_1^2}{2}$. The assumption $T\eta \geq \frac{16}{c_1} \log \frac{1}{\epsilon_{min}}$ in Lemma C.4 is satisfied by the definition of $T$.

The assumptions $(\tilde{\epsilon}_1^2 + r)C_x \leq \frac{c_1}{20}$ and $\frac{c_1^2}{8\eta\delta^2} \geq \log \frac{6rT^2}{\rho}$ in Lemma C.5 are satisfied by the same reason as that of Lemma C.2. The assumption $T\eta^2\delta^2 \geq 16 \log\left(\frac{6\sqrt{d}}{\rho\sqrt{\epsilon_1}}\right)$ in Lemma C.7 is satisfied by the definition of $T$, the assumption $\frac{2}{3}\eta\delta^2 > 32(\tilde{\epsilon}_1^2 + r)C_x b_x^2$ in Lemma C.7 is satisfied by the definition of $C_x$.

Since data are randomly from $\mathcal{N}(0, I)$, with $n \geq \widetilde{\Theta}((\frac{r \log(1/\epsilon_1)}{\rho})^2)$ data, there is with probability at least $1 - \frac{\rho}{18}$ there is $C_x b_x^2 (\tilde{\epsilon}_1^2 + r) \leq \min\left\{\frac{\eta\delta^2}{48}, \frac{\rho}{12T\eta}\right\} = \widetilde{\Theta}(\log(1/\epsilon_1)/\eta)$. Meanwhile, according to Lemma G.2, with $n \geq \widetilde{\Theta}(1)$ data with probability at least $1 - \frac{\rho}{18}$ there is $\mathbb{E}_i[(x_k^{(i)})^2] \geq \frac{2}{3}$ for all $k \in [d]$. According to definition of $b_x$, we know when $n \leq d$, with probability at least $1 - \frac{\rho}{18}$ there is also $\|x^{(i)}\|_\infty \leq b_x$ for all $i \in [n]$. In summary, with $d \geq n \geq \widetilde{\Theta}((\frac{r \log(1/\epsilon_1)}{\rho})^2)$ data, with probability at least $1 - \frac{\rho}{6}$ there is $C_x b_x^2 (\tilde{\epsilon}_1^2 + r) \leq \min\left\{\frac{\eta\delta^2}{48}, \frac{\rho}{12T\eta}\right\}$ and $\mathbb{E}_i[(x_k^{(i)})^2] \geq \frac{2}{3}$ for all $k \in [d]$ and $\|x^{(i)}\|_\infty \leq b_x$ for all $i \in [n]$.

Now we use these lemmas to finish the proof of the theorem. Let $\tilde{v}^{[t]}$ be a $(1 + c_1, \tilde{\epsilon}_1)$-bounded coupling of $v^{[t]}$, we only need to prove with probability at least $1 - \rho$, there is $\left\|\tilde{v}_S^{[T]} - 1\right\|_\infty \leq c_1$ and $\left\|\tilde{v}_{\bar{S}}^{[T]}\right\|_1 \leq \epsilon_1$, which follows from a union bound of the previous propositions. In particular, Lemma C.2 and Lemma C.3 tell us that probability of $\left\|\tilde{v}_S^{[T]}\right\|_\infty > 1 + c_1$ or $\left\|\tilde{v}_{\bar{S}}^{[T]}\right\|_1 > \tilde{\epsilon}_1$ is at most $\frac{\rho}{3}$. Lemma C.4 and Lemma C.5 tell us for any $k \in S$, probability of $\tilde{v}_k^{[T]} < 1 - c_1$ and $\left\|\tilde{v}_S^{[T]}\right\|_\infty \leq 1 + c_1$ and $\left\|\tilde{v}_{\bar{S}}^{[T]}\right\|_1 \leq \tilde{\epsilon}_1$ is at most $\frac{\rho}{3r}$. Lemma C.7 tells us the probability of $\left\|\tilde{v}_{\bar{S}}^{[T]}\right\|_1 > \epsilon_1$ and $\left\|\tilde{v}_S^{[T]}\right\|_\infty \leq 1 + c_1$ and $\left\|\tilde{v}_{\bar{S}}^{[T]}\right\|_1 \leq \tilde{\epsilon}_1$ is at most $\frac{\rho}{6}$. Combining them together tells us that probability of $\left\|\tilde{v}_S^{[T]} - 1\right\|_\infty > c_1$ or $\left\|\tilde{v}_{\bar{S}}^{[T]}\right\|_1 > \epsilon_1$ is at most $\rho$. $\qquad \square$

# D   PROOF OF CONVERGENCE TO GROUND TRUTH

The conclusion of Theorem 3.3 still allows constant error in the support, namely, $\|v_S - v_S^\star\|_\infty \leq 1/10$. The following theorem shows that further annealing the learning rate will let the algorithm fully converge to $v^\star$ with any target error. The end-to-end proof of the convergence to ground truth can be found in the next section (Section E).

**Theorem D.1.** *Let $s \geq 2$ be the index of the current round of bootstrapping. Let constant $c_0 = 1/10$. In the setting of Theorem 2.1, assume $v^{[0]}$ is an initial parameter satisfying $\|v_S^{[0]} - v_S^\star\|_\infty \leq c_{s-1}$ and $\|v_{\bar{S}}^{[0]} - v_{\bar{S}}^\star\|_1 \leq \epsilon_{s-1}$, where $0 < \epsilon_{s-1} \leq c_{s-1} \leq c_0$. Given a failure rate $\rho > 0$. Assume $n \geq \widetilde{\Theta}(r^2)$. Suppose we run SGD with label noise with noise level $\delta \geq 0$ and learning rate $\eta \leq \widetilde{\Theta}(c_s^2/(\delta^2 + r^2))$ for $T = \log(4/c_0)/\eta$ iterations. Then, with probability at least $1 - \rho$ over the randomness of the algorithm and data, there is $\|v_S^{[T]} - v_S^\star\|_\infty \leq c_s \triangleq c_{s-1}c_0$ and $\|v_{\bar{S}}^{[T]} - v_{\bar{S}}^\star\|_1 \leq \epsilon_s \triangleq (4/c_0)^{2c_{s-1}}\epsilon_{s-1}$. Here $\widetilde{\Theta}(\cdot)$ omits poly logarithmic dependency on $\rho$.*

In the rest of this section, we will first prove several lemmas on which the proof of Theorem D.1 is built upon. Then we will provide a proof of Theorem D.1.

**Definition D.2.** *($(b, \epsilon)$-to-$v^\star$ coupling) Let $v^{[0]}, v^{[1]}, \cdots, v^{[T]}$ be a trajectory of label noise gradient descent with initialization $v^{[0]}$. Recall $S \subset [d]$ is the support set of $v^\star$. We call the following random sequence $\tilde{v}^{[t]}$ a $(b, \epsilon)$-to-$v^\star$ coupling of $v^{[t]}$: starting from $\tilde{v}^{[0]} = v^{[0]}$, for each time $t < T$, if $\left\|\tilde{v}_{\bar{S}}^{[t]}\right\|_1 \leq \epsilon$ and $\left\|\tilde{v}_S^{[t]} - 1\right\|_\infty \leq b$, we let $\tilde{v}^{[t+1]} \triangleq v^{[t+1]}$; otherwise $\tilde{v}^{[t+1]} \triangleq \tilde{v}^{[t]}$.*

**Lemma D.3.** *In the setting of Theorem D.1, let $C_x \triangleq \max_{j \neq k} |\mathbb{E}_i[x_j^{(i)} x_k^{(i)}]|$. Assume $\left\|x^{(i)}\right\|_\infty \leq b_x$ for $i \in [n]$ for some $b_x > 0$, and $\mathbb{E}_i[(x_k^{(i)})^2] \geq \frac{2}{3}$ for $k \in [d]$. Let $\tilde{v}^{[t]}$ be a $(2c_{s-1}, \epsilon_s)$-to-$v^\star$ coupling of $v^{[t]}$. Assume $\frac{c_{s-1}^2}{2\eta b_x^2(\delta^2 + b_x^2(\epsilon_s^2 + r^2))} \geq \log \frac{10rT^2}{\rho}$ and $(\epsilon_s^2 + 4rc_{s-1})C_x b_x^2 \leq \frac{c_{s-1}}{10}$. Then, with probability at least $1 - \frac{\rho}{5}$, there is $\left\|\tilde{v}_S^{[T]} - 1\right\|_\infty \leq 2c_{s-1}$.*

*Proof of Lemma D.3.* For any fixed $1 \leq t_1 < t_2 \leq T$ and dimension $k \in S$, we consider the event that $\tilde{v}^{[t_1]} \in [1 + \frac{2}{3}c_{s-1}, 1 + c_{s-1}]$, and at time $t_2$ it is the first time in the trajectory such that $\tilde{v}^{[t_2]} > 1 + 2c_{s-1}$. We first bound the probability of this event happens, i.e., the following quantity:

$$\Pr\left(\tilde{v}_k^{[t_2]} - 1 > 2c_{s-1} \wedge \tilde{v}_k^{[t_1]} - 1 \leq c_{s-1} \wedge \tilde{v}_k^{[t_1:t_2]} \in [1 + \frac{2}{3}c_{s-1}, 1 + 2c_{s-1}]\right), \quad (93)$$

where $\tilde{v}_k^{[t_1:t_2]} \in [1 + \frac{2}{3}c_{s-1}, 1 + 2c_{s-1}]$ means that for all $t$ such that $t_1 \leq t < t_2$, there is $1 + \frac{2}{3}c_{s-1} \leq \tilde{v}_k^{[t]} \leq 1 + 2c_{s-1}$.

Notice that when $\left\|\tilde{v}_{\bar{S}}^{[t]}\right\|_1 \leq \epsilon_s$ and $\left\|\tilde{v}_S^{[t]} - 1\right\|_\infty \leq 2c_{s-1}$ and $\tilde{v}_k^{[t_1:t+1]} \in [1 + \frac{2}{3}c_{s-1}, 1 + 2c_{s-1}]$, there is

$$\mathbb{E}[\tilde{v}_k^{[t+1]} - 1] = \mathbb{E}_{s_t, i_t}\left[\left(1 + \eta s_t x_k^{i_t} - \eta(\tilde{v}^{[t]\odot 2} - v^{\star\odot 2})^\top x^{(i_t)} x_k^{(i_t)}\right)\tilde{v}_k^{[t]} - 1\right] \quad (94)$$

$$\leq (\tilde{v}_k^{[t]} - 1) - \frac{2}{3}\eta\tilde{v}_k^{[t]}(\tilde{v}_k^{[t]} + 1)(\tilde{v}_k^{[t]} - 1) + \eta(\epsilon_s^2 + 4rc_{s-1})C_x b_x^2 \tilde{v}_k^{[t]} \quad (95)$$

$$\leq (1 - \eta)(\tilde{v}_k^{[t]} - 1). \quad (96)$$

where the first inequality is because $\left\|\tilde{v}_{\bar{S}}^{[t]}\right\|_2^2 \leq \epsilon_s^2$ and properties of the data, the second inequality is because $(\epsilon_s^2 + 4rc_{s-1})b_x^2 C_x \leq \frac{c_{s-1}}{10}$. Also, we can bound the variance of this martingale as

$$\text{Var}\left[\tilde{v}_k^{[t+1]} - 1 \mid \tilde{v}^{[t]} - 1\right] = \text{Var}\left[\eta s_t x_k^{(i_t)}\tilde{v}_k^{[t]}\right] + \text{Var}\left[\eta((\tilde{v}^{[t]\odot 2} - v^{\star\odot 2})^\top x^{(i_t)})x_k^{(i_t)}\tilde{v}_k^{[t]}\right] \quad (97)$$

$$\leq (\eta\delta b_x(1 + 2c_{s-1}))^2 + \eta^2(\epsilon_s^2 + r)^2 b_x^4(1 + 2c_{s-1})^2 \quad (98)$$

$$\leq 4\eta^2 b_x^2(\delta^2 + b_x^2(\epsilon_s^2 + r)^2), \quad (99)$$

By Lemma G.3, we have

$$\Pr(\tilde{v}_k^{[t_2]} - 1 > 2c_{s-1} \wedge \tilde{v}_k^{[t_1]} - 1 \le c_{s-1} \wedge \tilde{v}_k^{[t_1:t_2]} \in [1 + \frac{2}{3}c_{s-1}, 1 + 2c_{s-1}]) \tag{100}$$

$$\le e^{\frac{-c_{s-1}^2}{2\eta^2 b_x^2 (\delta^2 + b_x^2 (\epsilon_s^2 + r)^2) \Sigma_{t=0}^{t_2 - t_1 - 1} (1 - \eta)^{2t}}} \tag{101}$$

$$\le e^{\frac{-c_{s-1}^2}{2\eta b_x^2 (\delta^2 + b_x^2 (\epsilon_s^2 + r)^2)}}, \tag{102}$$

where the first inequality is by Lemma G.3, the second inequality is by taking the sum of denominator.

Similarly, we bound

$$\Pr\left(1 - \tilde{v}_k^{[t_2]} > 2c_{s-1} \wedge 1 - \tilde{v}_k^{[t_1]} \le c_{s-1} \wedge \tilde{v}_k^{[t_1:t_2]} \in [1 - 2c_{s-1}, 1 - \frac{2}{3}c_{s-1}]\right). \tag{103}$$

Notice that when $\left\|\tilde{v}_{\bar{S}}^{[t]}\right\|_1 \le \epsilon_s$ and $\left\|\tilde{v}_S^{[t]} - 1\right\|_\infty \le 2c_{s-1}$ and $\tilde{v}_k^{[t_1:t+1]} \in [1 - 2c_{s-1}, 1 - \frac{2}{3}c_{s-1}]$, there is

$$\mathbb{E}[1 - \tilde{v}_k^{[t+1]}] = \mathbb{E}_{s_t, i_t}\left[1 - (1 + \eta s_t x_k^{i_t} - \eta(\tilde{v}^{[t]\odot 2} - v^{\star \odot 2})^\top x^{(i_t)}) x_k^{(i_t)}) \tilde{v}_k^{[t]}\right] \tag{104}$$

$$\le (1 - \tilde{v}_k^{[t]}) - \frac{2}{3}\eta \tilde{v}_k^{[t]}(\tilde{v}_k^{[t]} + 1)(1 - \tilde{v}_k^{[t]}) + \eta(\epsilon_s^2 + 4rc_{s-1})C_x b_x^2 \tilde{v}_k^{[t]} \tag{105}$$

$$\le (1 - \eta)(1 - \tilde{v}_k^{[t]}). \tag{106}$$

where the first inequality is because $\left\|\tilde{v}_{\bar{S}}^{[t]}\right\|_2^2 \le \epsilon_s^2$ and the properties of data, the second inequality is because $(\epsilon_s^2 + 4rc_{s-1})C_x b_x^2 \le \frac{c_{s-1}}{10}$. So

$$\Pr(1 - \tilde{v}_k^{[t_2]} > 2c_{s-1} \wedge 1 - \tilde{v}_k^{[t_1]} \le c_{s-1} \wedge \tilde{v}_k^{[t_1:t_2]} \in [1 - 2c_{s-1}, 1 - \frac{2}{3}c_{s-1}]) \tag{107}$$

$$\le e^{\frac{-c_{s-1}^2}{2\eta^2 b_x^2 (\delta^2 + b_x^2 (\epsilon_s^2 + r)^2) \Sigma_{t=0}^{t_2 - t_1 - 1} (1 - \eta)^{2t}}} \tag{108}$$

$$\le e^{\frac{-c_{s-1}^2}{2\eta b_x^2 (\delta^2 + b_x^2 (\epsilon_s^2 + r)^2)}}, \tag{109}$$

where the first inequality is by Lemma G.3, the second inequality is by taking the sum of denominator.

Finally, we finish the proof with a union bound. Since if $\left\|\tilde{v}_S^{[T]} - 1\right\|_\infty > 2c_{s-1}$, either event in Equation 93 or in Equation 103 has to happen for some $k \in S$ and $1 \le t_1 < t_2 \le T$, so we have

$$\Pr\left(\left\|\tilde{v}_S^{[T]} - 1\right\|_\infty > 2c_{s-1}\right) \tag{110}$$

$$\le \sum_{k \in S} \sum_{1 \le t_1 < t_2 \le T} \Pr\left(\tilde{v}_k^{[t_2]} - 1 > 2c_{s-1} \wedge \tilde{v}_k^{[t_1]} - 1 \le c_{s-1} \wedge \tilde{v}_k^{[t_1:t_2]} \in [1 + \frac{2}{3}c_{s-1}, 1 + 2c_{s-1}]\right) \tag{111}$$

$$+ \sum_{k \in S} \sum_{1 \le t_1 < t_2 \le T} \Pr\left(1 - \tilde{v}_k^{[t_2]} > 2c_{s-1} \wedge 1 - \tilde{v}_k^{[t_1]} \le c_{s-1} \wedge \tilde{v}_k^{[t_1:t_2]} \in [1 - 2c_{s-1}, 1 - \frac{2}{3}c_{s-1}]\right) \tag{112}$$

$$\le 2rT^2 e^{\frac{-c_{s-1}^2}{2\eta b_x^2 (\delta^2 + b_x^2 (\epsilon_s^2 + r)^2)}} \tag{113}$$

$$\le \frac{\rho}{5}, \tag{114}$$

where the first inequality is by union bound, the second inequality is by previous results, the third inequality is by assumption of this lemma. $\square$

**Lemma D.4.** *In the setting of Lemma D.3, assume* $(\epsilon_s^2 + 4c_{s-1}r)C_x b_x^2 \leq c_{s-1}$, $\epsilon_s > (1 + \eta c_{s-1})^T \epsilon_{s-1}$ *and* $\frac{((1+\eta c_{s-1})^{-T}\epsilon_s - \epsilon_{s-1})^2}{2T\eta^2 b_x^2(\delta^2 + b_x^2(\epsilon_s^2+r)^2)\epsilon_s^2} \geq \log\frac{5}{\rho}$ . *Then, with probability at least* $1 - \frac{\rho}{5}$*, there is* $\left\|\tilde{v}_{\bar{S}}^{[T]}\right\|_1 \leq \epsilon_s$.

*Proof of Lemma D.4.* When $\left\|\tilde{v}_S^{[t]} - 1\right\|_\infty \leq 2c_{s-1}$ and $\left\|\tilde{v}_{\bar{S}}^{[t]}\right\|_1 \leq \epsilon_s$, for any $k \notin S$, there is:

$$\mathbb{E}\left[\tilde{v}_k^{[t+1]}\right] = \tilde{v}_k^{[t]} - \eta\mathbb{E}_{i_t}\left[((\tilde{v}^{[t]\odot 2} - v^{\star\odot 2})^\top x^{(i_t)})x_k^{(i_t)}\tilde{v}_k^{[t]}\right] \tag{115}$$

$$\leq \tilde{v}_k^{[t]} + \eta(\left\|\tilde{v}_{\bar{S}}^{[t]\odot 2}\right\|_1 + 4c_{s-1}r)C_x b_x^2 \tilde{v}_k^{[t]} \tag{116}$$

$$\leq \tilde{v}_k^{[t]} + \eta(\epsilon_s^2 + 4c_{s-1}r)C_x b_x^2 \tilde{v}_k^{[t]} \tag{117}$$

$$\leq (1 + \eta c_{s-1})\tilde{v}_k^{[t]}. \tag{118}$$

where the first inequality is because we can bound the dimensions in $S$ and those not in $S$ with $\left\|\tilde{v}_{\bar{S}}^{[t]\odot 2}\right\|_1 C_x b_x^2$ and $4c_{s-1}rC_x b_x^2$ respectively, the second inequality is by $\left\|\tilde{v}_{\bar{S}}^{[t]}\right\|_2^2 \leq \left\|\tilde{v}_{\bar{S}}^{[t]}\right\|_1^2$, the third is because $(\epsilon_s^2 + 4c_{s-1}r)C_x b_x^2 \leq c_{s-1}$. Summing over all $k \notin S$ we have $\mathbb{E}[\left\|\tilde{v}_{\bar{S}}^{[t+1]}\right\|_1] \leq (1 + \eta c_{s-1})\left\|\tilde{v}_{\bar{S}}^{[t]}\right\|_1$. This bound is obviously also true when $\left\|\tilde{v}_S^{[t]} - 1\right\|_\infty > 2c_{s-1}$ or $\left\|\tilde{v}_{\bar{S}}^{[t]}\right\|_1 > \epsilon_s$, in which case $\tilde{v}^{[t+1]} = \tilde{v}^{[t]}$.

Therefore we know $(1 + \eta c_{s-1})^{-t}\left\|\tilde{v}^{[t]}\right\|_1$ is a supermartingale. Also notice $|\left\|\tilde{v}_{\bar{S}}^{[t+1]}\right\|_1 - \mathbb{E}[\left\|\tilde{v}_{\bar{S}}^{[t+1]}\right\|_1]| \leq \eta\delta\epsilon_s$, By Azuma Inequality,

$$\Pr\left(\left\|\tilde{v}_{\bar{S}}^{[T]}\right\|_1 > \epsilon_s\right) \tag{119}$$

$$\leq e^{-\frac{((1+\eta c_{s-1})^{-T}\epsilon_s - \epsilon_{s-1})^2}{2T\eta^2 b_x^2(\delta^2 + b_x^2(\epsilon_s^2+r)^2)\epsilon_s^2}}, \tag{120}$$

$$\leq \frac{\rho}{4}. \tag{121}$$

here we are using $\epsilon_s > (1 + \eta c_{s-1})^T \epsilon_{s-1}$ by assumption and the last step is by assumption. $\square$

**Lemma D.5.** *In the setting of Lemma D.3, assume* $(1 - \eta)^T 2c_{s-1} < \frac{c_s}{2}$, $\frac{c_{s-1}^2}{2\eta\delta^2} \geq \log\frac{5r}{\rho}$*, and* $(\epsilon_s^2 + 4c_{s-1}r)C_x b_x^2 \leq \frac{c_s}{10}$*. Then, for any* $k \in S$*, with probability at least* $1 - \frac{\rho}{5r}$*, either* $\min_{t\leq T}|\tilde{v}_k^{[t]} - 1| \geq \frac{c_s}{2}$*, or* $\left\|\tilde{v}_S^{[T]} - 1\right\|_\infty > 2c_{s-1}$*, or* $\left\|\tilde{v}_{\bar{S}}^{[T]}\right\|_1 > \epsilon_s$.

*Proof of Lemma D.5.* We first consider when $\tilde{v}_k^{[t]} \in [1 + \frac{c_s}{2}, 1 + 2c_{s-1}]$. For some $t < T_2$, if $\left\|\tilde{v}_S^{[t]} - 1\right\|_\infty \leq c_{s-1}$ and $\left\|\tilde{v}_{\bar{S}}^{[t]}\right\|_1 \leq \epsilon_s$, there is

$$\mathbb{E}[\tilde{v}_k^{[t+1]} - 1] \tag{122}$$

$$= \tilde{v}_k^{[t]} - \eta\mathbb{E}_{i_t}[((\tilde{v}^{[t]\odot 2} - v^{\star\odot 2})^\top x^{(i_t)})x_k^{(i_t)}]\tilde{v}_k^{[t]} - 1 \tag{123}$$

$$\leq (\tilde{v}_k^{[t]} - 1) - \frac{2}{3}\eta\tilde{v}_k^{[t]}(\tilde{v}_k^{[t]} + 1)(\tilde{v}_k^{[t]} - 1) + \eta(\epsilon_s^2 + 4c_{s-1}r)C_x b_x^2 v_k^{[t]} \tag{124}$$

$$\leq (1 - \eta)(v_k^{[t]} - 1). \tag{125}$$

Here the first inequality is by assumption, the second inequality is because $(\epsilon_s^2 + 4c_{s-1}r)C_x b_x^2 \leq \frac{1}{10}c_s$ and $c_{s-1} \leq \frac{1}{10}$.

We define the event $E_t$ as $\left\|\tilde{v}_S^{[t]} - 1\right\|_\infty > 2c_{s-1}$ or $\left\|\tilde{v}_{\bar{S}}^{[t]}\right\|_1 > \epsilon_s$. Since $(1-\eta)^{T_2} 2c_{s-1} < \frac{c_s}{2}$ by assumption, if $\tilde{v}_k^{[0]} \in [1 + \frac{c_s}{2}, 1 + 2c_{s-1}]$, by Lemma G.4 we know:

$$\Pr\left(\min_{t\leq T} \tilde{v}_k^{[t]} > 1 + \frac{c_s}{2} \wedge \left\|\tilde{v}_S^{[T]} - 1\right\|_\infty \leq 2c_{s-1} \wedge \left\|\tilde{v}_{\bar{S}}^{[T]}\right\|_1 \leq \epsilon_s\right) \tag{126}$$

$$\leq e^{-\frac{\left(\frac{1}{2}c_s(1-\eta)^{-T} - c_{s-1}\right)^2}{2\eta b_x^2(\delta^2 + b_x^2(\epsilon_s^2 + r)^2)}} \tag{127}$$

$$\leq e^{-\frac{c_{s-1}^2}{2\eta b_x^2(\delta^2 + b_x^2(\epsilon_s^2 + r)^2)}} \tag{128}$$

$$\leq \frac{\rho}{5r}, \tag{129}$$

where the second inequality is because of assumption.

Similarly, when $\tilde{v}_k^{[0]} \in [1 - 2c_{s-1}, 1 - \frac{c_s}{2}]$, there is

$$\Pr\left(\max_{t\leq T} \tilde{v}_k^{[t]} < 1 - \frac{c_s}{2} \wedge \left\|\tilde{v}_S^{[T_2]} - 1\right\|_\infty \leq 2c_{s-1} \wedge \left\|\tilde{v}_{\bar{S}}^{[T]}\right\|_1 \leq \epsilon_s\right) \tag{130}$$

$$\leq e^{-\frac{c_{s-1}^2}{2\eta b_x^2(\delta^2 + b_x^2(\epsilon_s^2 + r)^2)}} \tag{131}$$

$$\leq \frac{\rho}{5r}. \tag{132}$$

Since $|\tilde{v}_k^{[0]} - 1| \leq c_{s-1}$ by assumption of Theorem D.1, by bounding the probability for $\tilde{v}_k^{[0]} \in [1 + \frac{c_s}{2}, 1 + 2c_{s-1}]$ and $[1 - 2c_{s-1}, 1 - \frac{c_s}{2}]$ repectively we finished the proof. $\qquad\square$

**Lemma D.6.** *In the setting of Lemma D.3, assume* $\frac{c_s^2}{8\eta b_x^2(\delta^2 + b_x^2(\epsilon_s^2 + r)^2)} \geq \log\frac{5rT^2}{\rho}$ *and* $(\epsilon_s^2 + 2rc_s)C_x b_x^2 \leq \frac{c_s}{20}$. *Then, for any dimension* $k \in S$, *with probability at most* $\frac{\rho}{5r}$, *there is* $\min_{t\leq T} |\tilde{v}_k^{[t]} - 1| \leq \frac{1}{2}c_s$ *and* $|\tilde{v}_k^{[T]} - 1| > c_s$ *and* $\left\|\tilde{v}_S^{[T]} - 1\right\|_\infty \leq 2c_{s-1}$ *and* $\left\|\tilde{v}_{\bar{S}}^{[T]}\right\|_1 \leq \epsilon_s$.

*Proof of Lemma D.6.* For any fixed $1 \leq t_1 < t_2 \leq T$, we consider the event that $\tilde{v}^{[t_1]} \in [1 + \frac{1}{3}c_s, 1 + \frac{1}{2}c_s]$, and at time $t_2$ it is the first time in the trajectory such that $\tilde{v}^{[t_2]} > 1 + c_s$. We first bound the probability of this event happens, i.e., the following quantity:

$$\Pr\left(\tilde{v}_k^{[t_2]} - 1 > c_s \wedge \tilde{v}_k^{[t_1]} - 1 \leq \frac{1}{2}c_s \wedge \tilde{v}_k^{[t_1:t_2]} \in [1 + \frac{1}{3}c_s, 1 + c_s]\right). \tag{133}$$

Notice that when $\left\|\tilde{v}_{\bar{S}}^{[t]}\right\|_1 \leq \epsilon_s$ and $\left\|\tilde{v}_S^{[t]} - 1\right\|_\infty \leq 2c_{s-1}$ and $\tilde{v}_k^{[t_1:t+1]} \in [1 + \frac{1}{3}c_s, 1 + c_s]$, there is

$$\mathbb{E}[\tilde{v}_k^{[t+1]} - 1] = \mathbb{E}_{s_t, i_t}\left[(1 + \eta s_t x_k^{i_t} - \eta\mathbb{E}_{i_t}[(\tilde{v}^{[t]\odot2} - v^{\star\odot2})^\top x^{(i_t)})x_k^{(i_t)}])\tilde{v}_k^{[t]} - 1\right] \tag{134}$$

$$\leq (\tilde{v}_k^{[t]} - 1) - \frac{2}{3}\eta\tilde{v}_k^{[t]}(\tilde{v}_k^{[t]} + 1)(\tilde{v}_k^{[t]} - 1) + \eta(\epsilon_s^2 + 2rc_s)C_x b_x^2 \tilde{v}_k^{[t]} \tag{135}$$

$$\leq (1 - \eta)(\tilde{v}_k^{[t]} - 1). \tag{136}$$

where the first inequality is because $\left\|\tilde{v}_{\bar{S}}^{[t]}\right\|_2^2 \leq \epsilon_s^2$, the second inequality is because $(\epsilon_s^2 + 2rc_s)C_x b_x^2 \leq \frac{c_s}{20}$. Also, we can bound the variance of this martingale as

$$\mathrm{Var}\left[\tilde{v}_k^{[t+1]} - 1 \mid \tilde{v}^{[t]} - 1\right] = \mathrm{Var}\left[\eta s_t x_k^{(i_t)} \tilde{v}_k^{[t]}\right] + \mathrm{Var}\left[\eta((\tilde{v}^{[t]\odot2} - v^{\star\odot2})^\top x^{(i_t)})x_k^{(i_t)} \tilde{v}_k^{[t]}\right] \tag{137}$$

$$\leq (\eta\delta b_x(1 + c_s))^2 + \eta^2(\epsilon_s^2 + r)^2 b_x^4(1 + c_s)^2 \tag{138}$$

$$\leq 4\eta^2 b_x^2(\delta^2 + b_x^2(\epsilon_s^2 + r)^2), \tag{139}$$

By Lemma G.3, we have

$$\Pr(\tilde{v}_k^{[t_2]} - 1 > c_s \wedge \tilde{v}_k^{[t_1]} - 1 \le \frac{1}{2}c_s \wedge \tilde{v}_k^{[t_1:t_2]} \in [1 + \frac{1}{3}c_s, 1 + c_s]) \tag{140}$$

$$\le e^{\frac{-c_s^2}{8\eta^2 b_x^2(\delta^2 + b_x^2(\epsilon_s^2 + r)^2)\sum_{t=0}^{T-1}(1-\eta)^{2t}}} \tag{141}$$

$$\le e^{\frac{-c_s^2}{8\eta b_x^2(\delta^2 + b_x^2(\epsilon_s^2 + r)^2)}} \tag{142}$$

where the first inequality is by Lemma G.3, the second inequality is by taking the sum of denominator.

Similarly, we bound

$$\Pr(1 - \tilde{v}_k^{[t_2]} > c_s \wedge 1 - \tilde{v}_k^{[t_1]} \le \frac{1}{2}c_s \wedge \tilde{v}_k^{[t_1:t_2]} \in [1 - c_s, 1 - \frac{1}{3}c_s]) \tag{143}$$

$$\le e^{\frac{-c_s^2}{8\eta b_x^2(\delta^2 + b_x^2(\epsilon_s^2 + r)^2)}}. \tag{144}$$

Finally, we finish the proof with a union bound:

$$\Pr\left(\min_{t \le T} |\tilde{v}_k^{[t]} - 1| \le \frac{1}{2}c_s \wedge |\tilde{v}_k^{[T]} - 1| > c_s \wedge \left\|\tilde{v}_S^{[T]} - 1\right\|_\infty \le 2c_{s-1} \wedge \left\|\tilde{v}_{\bar{S}}^{[T]}\right\|_1 \le \epsilon_s\right) \tag{145}$$

$$\le \sum_{1 \le t_1 < t_2 \le T} \Pr(\tilde{v}_k^{[t_2]} - 1 > c_s \wedge \tilde{v}_k^{[t_1]} - 1 \le \frac{1}{2}c_s \wedge \tilde{v}_k^{[t_1:t_2]} \in [1 + \frac{1}{3}c_s, 1 + c_s]) \tag{146}$$

$$+ \sum_{1 \le t_1 < t_2 \le T} \Pr(1 - \tilde{v}_k^{[t_2]} > c_s \wedge 1 - \tilde{v}_k^{[t_1]} \le \frac{1}{2}c_s \wedge \tilde{v}_k^{[t_1:t_2]} \in [1 - c_s, 1 - \frac{1}{3}c_s]) \tag{147}$$

$$\le T^2 e^{\frac{-c_s^2}{8\eta b_x^2(\delta^2 + b_x^2(\epsilon_s^2 + r)^2)}} \tag{148}$$

$$\le \frac{\rho}{5r}, \tag{149}$$

where the first inequality is by union bound, the second inequality is by previous results, the third inequality is by assumption of this lemma. □

*Proof of Theorem D.1.* Let $C_x \triangleq \max_{j \ne k} |\mathbb{E}_i[x_j^{(i)} x_k^{(i)}]|$, $b_x = \sqrt{2 \log \frac{30d^2}{\rho}} = \widetilde{\Theta}(1)$. According to Lemma G.1, when $n \le d$, there is with probability at least $1 - \frac{\rho}{15}$ we have $\left\|x^{(i)}\right\|_\infty \le b_x$ for $i \in [d]$.

Set $\eta$ small enough such that $\frac{c_s^2}{8\eta b_x^2(\delta^2 + b_x^2(\epsilon_s^2 + r)^2)} \ge \log \frac{10rT^2}{\rho}$. Obviously we only need $\eta \le \widetilde{\Theta}(\frac{c_s^2}{\delta^2 + r^2})$, where $\widetilde{\Theta}(\cdot)$ omits poly logarithmic dependency on $d$ and $\rho$. Assume $(\epsilon_s^2 + 4c_{s-1}r)C_x b_x^2 \le \frac{c_s}{10}$, which can be represented as $C_x \le \widetilde{\Theta}(\frac{1}{r})$. Recall $T = \frac{1}{\eta} \log \frac{4}{c_0}$, $\epsilon_s = e^{2c_{s-1} \log \frac{4}{c_0}} \epsilon_{s-1}$.

We first show that the additional assumptions in the previous lemmas are satisfied. There is

$$(1 + \eta c_{s-1})^T = (1 + \eta c_{s-1})^{\frac{1}{\eta} \log \frac{4}{c_0}} \tag{150}$$

$$\le e^{c_{s-1} \log \frac{4}{c_0}} \triangleq P. \tag{151}$$

The assumption $\epsilon_s > (1 + \eta c_{s-1})^T \epsilon_{s-1}$ in Lemma D.4 is therefore satisfied by definition of $\epsilon_s$. The assumption $\frac{((1+\eta c_{s-1})^{-T}\epsilon_s - \epsilon_{s-1})^2}{2T\eta^2 b_x^2(\delta^2 + b_x^2(\epsilon_s^2 + r)^2)\epsilon_s^2} \ge \log \frac{5}{\rho}$ in Lemma D.4 is satisfied because:

$$\frac{((1 + \eta c_{s-1})^{-T}\epsilon_s - \epsilon_{s-1})^2}{2T\eta^2 b_x^2(\delta^2 + b_x^2(\epsilon_s^2 + r)^2)\epsilon_s^2} \ge \frac{\epsilon_s^2(P^{-1} - P^{-2})^2}{2T\eta^2 b_x^2(\delta^2 + b_x^2(\epsilon_s^2 + r)^2)\epsilon_s^2} \tag{152}$$

$$\ge \frac{(P - 1)^2}{2T\eta^2 b_x^2(\delta^2 + b_x^2(\epsilon_s^2 + r)^2)} \tag{153}$$

$$\ge \frac{c_{s-1}^2}{2\eta b_x^2(\delta^2 + b_x^2(\epsilon_s^2 + r)^2)}, \tag{154}$$

which is larger than $\log \frac{5}{\rho}$ by the definition of $\eta$. The assumption $(1-\eta)^T 2c_{s-1} \leq \frac{c_s}{2}$ in Lemma D.5 is satisfied because $(1-\eta)^T 2c_{s-1} \leq (\frac{1}{e})^{\log \frac{4}{c_0}} 2c_{s-1} = \frac{c_s}{2}$. All the other assumptions in Lemma D.3, Lemma D.4, Lemma D.5and Lemma D.6 naturally follows from the definition of $\eta$ and the requirement of $C_x$.

Since data are randomly from $\mathcal{N}(0, I)$, with $n \geq \widetilde{\Theta}(r^2)$ data, there is with probability at least $1 - \frac{\rho}{15}$ there is $(\epsilon_s^2 + 4c_{s-1}r)C_x b_x^2 \leq \frac{c_s}{10}$. Meanwhile, according to Lemma G.2 with $n \geq \widetilde{\Theta}(1)$ data with probability at least $1 - \frac{\rho}{15}$ there is $\mathbb{E}_i[(x_k^{(i)})^2] \geq \frac{2}{3}$ for all $k \in [d]$. According to definition of $b_x$, we know when $n \leq d$, with probability at least $1 - \frac{\rho}{15}$ there is also $\left\| x^{(i)} \right\|_\infty \leq b_x$ for all $i \in [n]$. In summary, with $d \geq n \geq \widetilde{\Theta}(r^2)$ data, with probability at least $1 - \frac{\rho}{5}$ there is $(\epsilon_s^2 + 4c_{s-1}r)C_x b_x^2 \leq \frac{c_s}{10}$ and $\mathbb{E}_i[(x_k^{(i)})^2] \geq \frac{2}{3}$ for all $k \in [d]$ and $\left\| x^{(i)} \right\|_\infty \leq b_x$ for all $i \in [n]$.

Now we finish the proof with the above lemmas. Lemma D.3 and Lemma D.4 together tell us that with probability at least $1 - \frac{2\rho}{5}$, there is $\left\| \tilde{v}_S^{[T]} - 1 \right\|_\infty \leq 2c_{s-1}$ and $\left\| \tilde{v}_{\bar{S}}^{[T]} \right\|_1 \leq \epsilon_s$, in which case there is also $v^{[T]} = \tilde{v}^{[T]}$ by the definition of $\tilde{v}^{[T]}$. Lemma D.5 and Lemma D.6 together tell us the probability of $\left\| \tilde{v}_S^{[T]} - 1 \right\|_\infty \leq 2c_{s-1}$ and $\left\| \tilde{v}_{\bar{S}}^{[T]} \right\|_1 \leq \epsilon_s$ and $\left\| \tilde{v}_S^{[T]} - 1 \right\|_\infty > c_s$ is no more than $\frac{2\rho}{5}$. So together we know with probability at least $1 - \rho$, there is $\left\| v_S^{[T]} - 1 \right\|_\infty \leq c_s$ and $\left\| v_{\bar{S}}^{[T]} \right\|_1 \leq \epsilon_s$. $\qquad \square$

# E  PROOF OF THEOREM 2.1

*Proof of Theorem 2.1.* Starting from initialization $\tau \cdot \mathbb{1}$, by Theorem 3.1, running SGD with label noise with noise level $\delta > \widetilde{O}(\frac{\tau^2 d^2}{\rho^3})$ and $\eta_0 = \widetilde{\Theta}(\frac{1}{\delta})$ for $T_0 = \widetilde{\Theta}(1)$ iterations gives us that with probability at least $0.99$, $\epsilon_{min} \leq v_k^{[T_0]} \leq \frac{1}{d}$ where $\epsilon_{min} = exp(-\widetilde{\Theta}(1))$. Now $v^{[T_0]}$ satisfies the initial condition of Theorem 3.3.

Recall the final target precision is $\epsilon$, set $\epsilon_1 = \frac{1}{40^3}\epsilon$. By Theorem 3.3, with $n \geq \widetilde{\Theta}(r^2 \log^2(1/\epsilon))$ data, after running SGD with label noise with learning rate $\eta_1 = \widetilde{\Theta}(\frac{1}{\delta^2})$ for $T_1 = \widetilde{\Theta}(\frac{\log(1/\epsilon)}{\eta_1})$ iterations, with probability at least $0.99$, there is,

$$\left\| v_S^{[T_0+T_1]} - v_S^{\star} \right\|_\infty \leq c_1 \triangleq \frac{1}{10}, \tag{155}$$

and

$$\left\| v_{\bar{S}}^{[T_0+T_1]} - v_{\bar{S}}^{\star} \right\|_1 \leq \epsilon_1. \tag{156}$$

So we have $v^{[T_0+T_1]}$ satisfies the initial condition of Theorem D.1.

Finally, set $\rho = 0.01/\lceil \log_{10}(1/\epsilon) \rceil$, and apply Theorem D.1 for $n_s = \lceil \log_{10}(1/\epsilon) \rceil = \widetilde{\Theta}(1)$ rounds. Since $c_s$ gets smaller by $1/10$ for each round, the final $c_{n_s}$ satisfies $\frac{1}{10}\epsilon \leq c_{n_s} \leq \epsilon$. Since the requirement of $\eta$ for round $s$ is $\eta \leq \widetilde{\Theta}(\frac{c_s^2}{\delta^2+r^2})$, we can set $\eta_2 \leq \widetilde{\Theta}(\frac{\epsilon^2}{\delta^2})$ to satisfy all the rounds at the same time. Set $T_2$ be the total number of iterations in all of these rounds, obviously $T_2 = \widetilde{\Theta}(\frac{1}{\eta_2})$. Notice that $\epsilon_s \leq e^{\sum_{s=2}^{\infty} 2c_s-1 \log \frac{4}{c_0}}\epsilon_1 \leq 40^3\epsilon_1 = \epsilon$, we have with probability at least $0.99$,

$$\left\| v_S^{[T_0+T_1+T_2]} - v_S^{\star} \right\|_\infty \leq c_1 \triangleq \epsilon, \tag{157}$$

and

$$\left\| v_{\bar{S}}^{[T_0+T_1+T_2]} - v_{\bar{S}}^{\star} \right\|_1 \leq \epsilon. \tag{158}$$

The total failure rate of above three stages is $0.03$, so with probability at least $0.97$, there is $\left\| v^{[T_0+T_1+T_2]} - v^{\star} \right\|_\infty \leq \epsilon$, which finishes the proof. $\qquad\square$

# F    PROOF OF THEOREM 2.2

**Lemma F.1.** *Assume $n \leq \frac{d}{2} - 9\sqrt{d}$. Let $C \subset \mathbb{R}^d$ be the convex cone where each coordinate is positive, $K$ be a random subspace of dimension $d - n$. Then with probability at least $0.999$, there is $K \cap C \neq \{0\}$*

*Proof of Lemma F.1.*  By Theorem 1 of Amelunxen et al. (2014), we only need to prove

$$\delta(C) + \delta(K) \geq d + 9\sqrt{d}, \tag{159}$$

where $\delta(\cdot)$ is the statistical dimension of a set. By equation (2.1) of Amelunxen et al. (2014), there is

$$\delta(K) = d - n. \tag{160}$$

To calculate $\delta(C)$, we use Proposition 2.4 from Amelunxen et al. (2014),

$$\delta(C) = \mathbb{E}[\|\Pi_C(g)\|^2], \tag{161}$$

where $g$ is a standard random vector, $\Pi_C$ is projection of $g$ to $C$, the expectation is over $g$. Since $C$ is the set of all points with element-wise positive coordinate, $\Pi_C(g)$ is simply setting all the negative dimension of $g$ to $0$ and keep the positive ones. Therefore,

$$\delta(C) = \mathbb{E}[\|\Pi_C(g)\|^2] = \frac{d}{2}. \tag{162}$$

Therefore we have

$$\delta(C) + \delta(K) = \frac{3}{2}d - n \geq d + 9\sqrt{d}. \tag{163}$$

$\square$

*Proof of Theorem 2.2.*  Let $X^\perp$ be the subspace that is orthogonal to the subspace $X$ spanned by data. Since data is random, with probability 1 the random subspace $X$ is of $n$ dimension. Therefore, according to the previous lemma, with probability at least $0.999$, there is $X^\perp \cap C \neq \{0\}$, where $C$ is the coordinate-wise positive cone. Let $\mu \in X^\perp$ be such a vector such that $\mu_i > 0$ for $\forall i \in [d]$, and we scale it such that $\|\mu\|_2 = 1$. We can construct the following orthonormal matrix

$$A = [a^1, \cdots, a^d] \in \mathbb{R}^{d \times d}, \tag{164}$$

such that $span\{a^1, \cdots a^n\} = X$ and $a^{n+1} = \mu$. Consider the following transformation

$$A\tilde{u} = u = v^{\odot 2}, \tag{165}$$

since only the projection of $u$ to the span of data influences $L(v)$, we can write $L(v) = \tilde{L}(\tilde{u}_{1:n})$ as a function of the first $n$ dimensions of $\tilde{u}$.

We can lower bound the partition function with

$$\int_{v \in \mathbb{R}^d} e^{-\lambda L(v)} dv \geq \int_{v > 0} e^{-\lambda L(v)} dv \tag{166}$$

$$= \int_{A\tilde{u} > 0} e^{-\lambda \tilde{L}(\tilde{u}_{1:n})} \det \frac{\partial v}{\partial u} \det \frac{\partial u}{\partial \tilde{u}} d\tilde{u} \tag{167}$$

$$= \frac{1}{2^d} \int e^{-\lambda \tilde{L}(\tilde{u}_{1:n})} \left( \int_{A\tilde{u} > 0} \prod_{i=1}^d \frac{1}{\sqrt{u_i}} d\tilde{u}_{n+1:d} \right) d\tilde{u}_{1:n}. \tag{168}$$

Here the inner loop is integrating over the last $d - n$ dimensions of $\tilde{u}$ in the set such that $A\tilde{u}$ is coordinate-wise positive. Now we prove that for each $\tilde{u}_{1:n}$ such that $S = \{\tilde{u}_{n+1;d} | A\tilde{u} > 0\}$ is not empty set, the inner loop integral is always $+\infty$.

Fix $\tilde{u}_{1:n}$, let $\tilde{u}^*_{n+1:d}$ be one possible solution such that $u^* = A\tilde{u} > 0$. Define constant

$$c = \min_{i \leq [1:d]} \max_{j \in [n+2:d]} \frac{a_i^{n+1}}{(d - n - 1)|a_i^j|}, \tag{169}$$

we can define the following set

$$S' = \{\tilde{u}_{n+1:d} | \tilde{u}_{n+1} \geq \tilde{u}_{n+1}^* \land |\tilde{u}_j - \tilde{u}_j^*| \leq c(\tilde{u}_{n+1} - \tilde{u}_{n+1}^*), \forall j \in [n+2, d]\} \tag{170}$$

In other words, this is a convex cone where constraint of $\tilde{u}_j$ is linear in $\tilde{u}_{n+1}$ for $j \in [n+2 : d]$. By definition of $c$, it is easy to verify that $S'$ is a subset of $S$. Also, for every $\tilde{u}_{n+1:d} \in S'$, $u_i$ is upper bounded by

$$\left(A \begin{bmatrix} \tilde{u}_{1:n} \\ \tilde{u}_{n+1:d} \end{bmatrix}\right)_i = u_i^* + a_i^{n+1}(\tilde{u}_{n+1} - \tilde{u}_{n+1}^*) + \sum_{j=n+2}^{d} a_i^j(\tilde{u}_j - \tilde{u}_j^*) \tag{171}$$

$$\leq u_i^* + 2a_i^{n+1}(\tilde{u}_{n+1} - \tilde{u}_{n+1}^*). \tag{172}$$

Here the inequality is because of the definition of $c$.

Let $z = \tilde{u}_{n+1} - \tilde{u}_{n+1}^*$ we have

$$\int_{\tilde{u}_{n+1:d} \in S'} \prod_{i=1}^{d} \frac{1}{\sqrt{u_i}} d\tilde{u}_{n+1:d} \tag{173}$$

$$\geq \int_{z \geq 0} (2cz)^{d-n-1} \prod_{i=1}^{d} \frac{1}{\sqrt{u_i^* + 2a_i^{n+1}z}} dz \tag{174}$$

$$= +\infty. \tag{175}$$

Here the last step is because $n < d/2$, so the integrand is essentially a polynomial of $z$ with degree larger than $-1$, so integrating it over all positive $z$ has to be $+\infty$. So we finish the proof that $\int_{v \in \mathbb{R}^d} e^{-\lambda L(v)} dv = +\infty$. $\qquad \square$

## G    EXTRA LEMMAS

**Lemma G.1.** *Suppose $x^{(i)} \sim \mathcal{N}(0, \mathcal{I}_{d \times d})$ where $i \in [n]$ are random data. Then with probability at least $1 - \rho$, for every $i \in [n]$ there is*

$$\left\| x^{(i)} \right\|_\infty \leq \sqrt{2 \log \frac{2nd}{\rho}} \tag{176}$$

*Proof.* By Gaussian tail bound, there is $\Pr\left( |x_k^{(i)}| > b_x \right) \leq 2e^{-\frac{b_x^2}{2}}$. So by union bound we have $\Pr\left( \max_{i,k} |x_k^{(i)}| > b_x \right) \leq 2nd e^{-\frac{b_x^2}{2}}$. Let $b_x = \sqrt{2 \log \frac{2nd}{\rho}}$ we finished the proof. $\qquad \square$

**Lemma G.2.** *Suppose $x^{(i)} \sim \mathcal{N}(0, \mathcal{I}_{d \times d})$ where $i \in [n]$ are random data. Then when $n > 24 \log \frac{d}{\rho}$, with probability at least $1 - \rho$, for every $k \in [d]$ there is*

$$\mathbb{E}_i[x_k^{(i)^2}] \geq \frac{2}{3}. \tag{177}$$

*Proof.* Since $\mathbb{E}_x[{x_k}^2] = 1$, $\mathbb{E}_x[{x_k}^4] = 3$, by Hoeffding inequality we have

$$\Pr\left( \frac{1}{n} \sum_{i=1}^n x_k^{(i)^2} < \frac{2}{3} \right) \leq e^{\frac{-n}{24}}. \tag{178}$$

Therefore, when $n \geq 24 \log \frac{d}{\rho}$, by union bound we finish the proof. $\qquad \square$

**Lemma G.3.** *Let $c > 0$, $1 > \gamma > 0$ be real constants. Let $A^{[0]}, A^{[1]}, \cdots, A^{[T]}$, be a series of random variables, such that given $A^{[0]}, \cdots, A^{[t]}$ for some $t < T$ with $A^{[t]} \in [\frac{c}{3}, c]$, there is either $A^{[t]} = A^{[t+1]} = \cdots = A^{[T]}$, or $\mathbb{E}[A^{[t+1]}] \leq (1 - \gamma)A^{[t]}$ with variance $\mathrm{Var}[A^{[t+1]} \mid A^{[0]}, \cdots, A^{[t]}] \leq a$. Then there is*

$$\Pr\left( A^{[T]} > c \wedge A^{[0]} \leq \frac{c}{2} \wedge A^{[0:T]} \in \left[ \frac{c}{3}, c \right] \right) \leq e^{\frac{-c^2}{2a \sum_{t=0}^{T-1} (1-\gamma)^{2t}}}. \tag{179}$$

*where $A^{[0:T]} \in \left[ \frac{c}{3}, c \right]$ means for any $0 \leq t < T$, $A^{[t]} \in \left[ \frac{c}{3}, c \right]$.*

*Proof of Lemma G.3.* We only need to consider when $A^{[0]} \leq \frac{c}{2}$. Let $\hat{A}^{[t]}$ be the following coupling of $A^{[t]}$: starting from $\hat{A}^{[0]} = A^{[0]}$, for each time $t < T$, if $A^{[t]} = A^{[t+1]} = \cdots = A^{[T]}$ or there is $t' \leq t$ such that $A^{[t']} \notin [\frac{c}{3}, c]$, we let $\hat{A}^{[t+1]} \triangleq (1 - \gamma)\hat{A}^{[t+1]}$; otherwise $\hat{A}^{[t+1]} = A^{[t+1]}$. Intuitively, whenever $A^{[t]}$ stops updating or exceeds proper range, we only times $\hat{A}^{[t]}$ by $1 - \gamma$ afterwards, otherwise we let it be the same as $A^{[t]}$. Notice that if the event in Equation 179 happens, there has to be $\hat{A}^{[T]} = A^{[T]}$ (otherwise $A^{[t]}$ stops updating or exceeds range at some time, contradicting the event). So we only need to bound $\Pr\left( \hat{A}^{[T]} > c \right)$.

We notice that $(1 - \gamma)^{-t}\hat{A}^{[t]}$ for $t = 0 \cdots T$ is a supermartingale, i.e., given history there is $\mathbb{E}[\hat{A}^{[t+1]}|\hat{A}^{[t]}] \leq (1 - \gamma)\hat{A}^{[t]}$. This is obviously true when $\hat{v}^{[t+1]} = (1 - \gamma)\hat{A}^{[t]}$, and also true otherwise by assumption of the lemma. So we have

$$\Pr(\hat{A}^{[T]} > c) \tag{180}$$

$$= \Pr((1 - \gamma)^{-T}\hat{A}^{[T]} > (1 - \gamma)^{-T}c) \tag{181}$$

$$\leq e^{-\frac{2\left( c(1-\gamma)^{-T} - \hat{A}^{[0]} \right)^2}{\sum_{t=0}^{T-1} (1-\gamma)^{-2t} a}} \tag{182}$$

$$\leq e^{-\frac{c^2 (1-\gamma)^{-2T}}{2 \sum_{t=0}^{T-1} (1-\gamma)^{-2(t+1)} a}} \tag{183}$$

$$= e^{-\frac{c^2}{2a \sum_{t=0}^{T-1} (1-\gamma)^{2t}}}. \tag{184}$$

where the first inequality is because of Azuma Inequality and $\mathrm{Var}\left[(1-\gamma)^{-t+1}\hat{A}^{[t+1]} \mid \hat{A}^{[t]}\right] \leq$ $(1-\gamma)^{-2(t+1)}a.$ , the second inequality is because $\hat{A}^{[0]} \leq \frac{c}{2}$. Since the event in Equation 179 only happens when $\hat{A}^{[T]} > c$, we've finished the proof. □

**Lemma G.4.** *Let $0 < c_1 < c_2$ be real constants. Let $A^{[0]}, A^{[1]}, \cdots, A^{[T]}$, be a series of random variables, such that given $A^{[0]}, \cdots, A^{[t]}$ for some $t < T$ with $A^{[t]} \in [c_1, c_2]$, there is either event $E_t$ happens, or $\mathbb{E}[A^{[t+1]}] \leq (1-\gamma)A^{[t]}$ with variance $\mathrm{Var}[A^{[t+1]} \mid A^{[0]}, \cdots, A^{[t]}] \leq a$. Then when $A^{[0]} \in [c_1, c_2]$ and $(1-\gamma)^T c_2 < c_1$ there is*

$$\Pr\left(\min_{t \leq T} A^{[t]} > c_1 \wedge \max_{t \leq T} A^{[t]} \leq c_2 \wedge \neg E_{[0:T]}\right) \leq e^{-\frac{2\left(c_1(1-\gamma)^{-T}-A^{[0]}\right)^2}{\frac{1}{\gamma}a}}. \tag{185}$$

*where $\neg E_{[0:T]}$ means for any $0 \leq t < T$, $E_t$ doesn't happen.*

*Proof of Lemma G.4.* Let $\hat{A}^{[t]}$ be the following coupling of $A^{[t]}$: starting from $\hat{A}^{[0]} = A^{[0]}$, for each time $t < T$, if exists $t' \leq t$ such that $E_{t'}$ happens or $A^{[t']} \notin [c_1, c_2]$, we let $\hat{A}^{[t+1]} \triangleq (1-\gamma)\hat{A}^{[t+1]}$; otherwise $\hat{A}^{[t+1]} = A^{[t+1]}$. Intuitively, whenever $A^{[t]}$ exceeds proper range, we only times $\hat{A}^{[t]}$ by $1-\gamma$ afterwards, otherwise we let it be the same as $A^{[t]}$. Notice that if the event in Equation 185 happens, there has to be $\hat{A}^{[T]} = A^{[T]}$ (otherwise $E_t$ happens sometimes or $A^{[t]} \notin [c_1, c_2]$, contradicting the event). So we only need to bound $\Pr\left(\hat{A}^{[T]} > c_1\right)$.

We notice that $(1-\gamma)^{-t}\hat{A}^{[t]}$ for $t = 0 \cdots T$ is a supermartingale, i.e., given history there is $\mathbb{E}[\hat{A}^{[t+1]}|\hat{A}^{[t]}] \leq (1-\gamma)\hat{A}^{[t]}$. This is obviously true when $\hat{v}^{[t+1]} = (1-\gamma)\hat{A}^{[t]}$, and also true otherwise by assumption of the lemma. So we have

$$\Pr(\hat{A}^{[T]} > c_1) \tag{186}$$

$$= \Pr((1-\gamma)^{-T}\hat{A}^{[T]} > (1-\gamma)^{-T}c_1) \tag{187}$$

$$\leq e^{-\frac{2\left(c_1(1-\gamma)^{-T}-A^{[0]}\right)^2}{\Sigma_{t=0}^{T-1}(1-\gamma)^{-2t}a}} \tag{188}$$

$$\leq e^{-\frac{2\left(c_1(1-\gamma)^{-T}-A^{[0]}\right)^2}{\frac{1}{\gamma}a}} \tag{189}$$

where the first inequality is because of Azuma Inequality and $\mathrm{Var}\left[(1-\gamma)^{-t+1}\hat{A}^{[t+1]} \mid \hat{A}^{[t]}\right] \leq$ $(1-\gamma)^{-2(t+1)}a$ and $\hat{A}^{[0]} \leq c_2 \leq e^{-\frac{2\left(c_1(1-\gamma)^{-T}-c_2\right)^2}{\Sigma_{t=0}^{T-1}(1-\gamma)^{-2t}a}}$. Since the event in Equation 185 only happens when $\hat{A}^{[T]} > c_1$, we've finished the proof. □

