# OpenReview forum: "Shape Matters: Understanding the Implicit Bias of the Noise Covariance"
_ICLR.cc/2021/Conference — Reject_

### Official Review · AnonReviewer1 · 2020-10-28
**Solid contribution with theoretical insights on SGD with label noise**

**Rating:** 7
**Confidence:** 3

**Review:**

This paper considers the implicit regularization of stochastic gradient decent (SGD). The authors analyze SGD with label nose in the quadratically-parameterized model and prove that it converges to the sparse ground-truth even if started with large initialization. The authors also prove that SGD with Gaussian noise (Langevin dynamics) does not converge to the ground truth at zero under the overparameterized regime.

This is a solid contribution with theoretical insights on SGD with label noise. While the theoretical results are deep with long proofs, their outlines and meanings are well explained.

- The difference (affinity) between the deep neural network and the quadratically-parameterized model is mainly discussed numerically (Figure 1). It would be nicer to discuss gaps between them theoretically, if possible. For example, the contraction of SGD in the initial phase (Thm 3.1) is reminiscent of the effect of singularities in neural networks discussed in references such as:
Guo et al.: Numerical Analysis near Singularities in RBF networks, JMLR, 19(2018), 1-39.
It would be nicer to discuss if this analysis of the initial phase has something to do with singularities.

- p.7, footnote 7: What is the second-order effect (of zero-mean noise)?

Minor:
p.4, Langevin dynamics/diffusion: The last sentence is duplicated.

---

> ### Author Response · Authors · 2020-11-15
> **Response**
>
> We thank the reviewer for the comments and feedback! We will incorporate feedback and comments into the next version of our paper.
>
> Responses to specific points below:
>
>
>
> *-- “It would be nicer to discuss if this analysis of the initial phase has something to do with singularities.”*
>
> We thank the reviewer for pointing out the potential connection between our theory and the idea of singularities in [Guo et al. 2018]. Our Theorem 3.1 shows that GD with label noise converges to parameters with small norm, which helps achieve better generalization. In comparison, [Guo et al. 2018] shows that for a RBF network, GD can sometimes be trapped around some parameter regime with small norm (termed “elimination singularity”).
>
> Conceptually, both our theory and the singularities theory involves a regime where the parameter has a small norm. However, the effect of the small norm is different: in our work the small norm is beneficial to recovering the ground truth, while in [Guo et al. 2018] small norm parameters make the optimization slower. One interesting future direction might be the potential tradeoff between better generalization and faster optimization. We have added [Guo et al. 2018] and the above connection to our related works section.
>
> *-- “What is the second-order effect (of zero-mean noise) on a potential function?”*
>
> The “second-order effect” refers to the noise’s influence on the expectation of potential function under second-order Taylor approximation (see the last equation in Section 3.1, page 6). In general for any potential function on model parameters, adding zero-mean noise to parameters won’t change the expectation of potential function under first order Taylor approximation. However, when the Hessian of the potential function is non-zero, adding zero-mean noise will change the expectation of potential function under second-order Taylor approximation.
>
> This second-order effect is an essential step in the proof of Theorem 3.1. Concretely, we define a concave potential function $\phi(\cdot)$ and show that it decreases in expectation due to the second-order effect. The decreasing potential function is further translated into the convergence of parameters.
>
> *-- “The difference (affinity) between the deep neural network and the quadratically-parameterized model is mainly discussed numerically (Figure 1). It would be nicer to discuss gaps between them theoretically, if possible.”*
>
> We suspect that our high level intuition --- training with state-dependent noise follows a path with smaller noise --- may still hold for neural networks. However, to formally generalize our theory to neural nets, the fundamental obstacle is that we lack sufficient structures to analyze the optimization dynamics for a non-convex function.  The direct technical challenge is how to properly choose a potential function for which we can control iteratively. Analyzing  the global dynamics of nonconvex optimization for neural nets---with the regularization effect being taken into account --- appears to be a major open question in the area of deep learning theory.
>
> References:
>
> [Guo et al. 2018] Guo, W., Wei, H., Ong, Y. S., Hervas, J. R., Zhao, J., Wang, H., & Zhang, K. (2018). Numerical analysis near singularities in RBF networks. The Journal of Machine Learning Research, 19(1), 1-39.

---

### Official Review · AnonReviewer3 · 2020-10-29
**the authors study an interesting problem and provide sound theoretical analysis**

**Rating:** 6
**Confidence:** 3

**Review:**

The authors study the problem quadratically parameterized (linear) regression and study the behavior of SGD to solve it when the stochasticity added is in terms of label noise. They show that SGD (with this kind of noise) with arbitrarily large initialization converges to the ground truth solution, whereas there exist settings where Langevin dynamics or gradient descent would fail to converge to this solution. The proofs are carried out carefully and are correct as best as I could verify. The authors also provide experiments on synthetic data to support and motivate their theory.


Some questions/comments:
1. Why are three stages with decreasing step-sizes needed for the analysis? Can we expect a similar result to hold if a constant, but, small step-size is chosen instead?
2. The generative model assumes that the label y has no added noise. Are the theoretical result robust to additive noise?
3. The authors point out on page 4 that the sample complexity is worse that LASSO (which is fine), but, do not remark why this is the case. It would be insightful if the authors could add a comment about why this is the case, and if this was experimentally observed as well.

---

> ### Author Response · Authors · 2020-11-15
> **Response**
>
> We thank the reviewer for the comments and insightful questions!
>
> Responses to specific points below:
>
> *-- “Why are three stages with decreasing step-sizes needed for the analysis? Can we expect a similar result to hold if a constant, but, small step-size is chosen instead?”*
>
> We study three stages of decreasing step-sizes since the analysis can be simpler and more straightforward in this setting. Moreover, empirically, for the quadratically-parameterized model (and real datasets), though a small constant step-size can also work, its convergence is slower than decaying lr schedule. Thus, we chose to mostly focus on the decaying lr schedule.
>
> Concretely,  our current analysis shows a clear contrast between stages --- the effect of noise dominates the gradient in the first stage (and the second stage for sparse dimensions), vice versa in the last stage (and the second stage for non-sparse dimensions). Analyzing constant step-size is interesting and more challenging and is left for future work due to the more complicated interaction between noise and gradient.
>
> *-- “The generative model assumes that the label y has no added noise. Are the theoretical results robust to additive noise?”*
>
> Yes, our theory can be generalized (straightforwardly) to the setting where the label has additive noise, though in this case we can only recover the ground truth approximately because the noise precludes exact recovery. We would also like to clarify that even with the intrinsic noise in the label, additional artificial noise in the optimizer is still needed to achieve good generalization, both empirically and theoretically. This suggests that the artificial noise is more fundamental to the improved generalization.
>
> We will add this extension in the next revision. We primarily focused on artificial noise in the optimization procedure because we aimed to study the  implicit bias of the optimizers
>
> *-- “The authors point out on page 4 that the sample complexity is worse than LASSO (which is fine), but, do not remark why this is the case. It would be insightful if the authors could add a comment about why this is the case.”*
>
> As we briefly mentioned in the footnote 3, this is due to the technical limitation of analyzing nonconvex optimization. Most of the nonconvex optimization algorithms (even without overparameterization and implicit regularization) do not have the optimal sample complexity guarantees in the sparsity as the convex relaxations can. E.g., the works on nonconvex optimization for matrix completion or matrix sensing often suffers from a suboptimal dependency on the rank r ([Li et al. 2018], [Ge et al. 2016] and [Chi et al. 2019]), where [Ge et al. 2016] and [Chi et al. 2019] don’t consider any overparameterization. Our direct prior work [Vaskevicius et al. 2019] also had the same limitations. We didn’t focus on getting the optimal dependency in r because the main message is that we can get much better dependency in d, compared to GD.
>
> More concretely, in both our analysis and the proof of LASSO’s bound, one important step is to show that with high probability, the data matrix $E_i [x_i x_i^\top]$ satisfies some property. For the proof of LASSO, the property that we need is the restricted eigenvalue of $E_i [x_i x_i^\top]$ in some convex cone to be strictly positive (see page 295 of [Hastie et al. 2015]). However, our analysis requires that each off-diagonal element of the matrix to be smaller than O(1/r), which is a stronger assumption and hence can be guaranteed only with more data.  We believe this is fundamentally the same technical challenge showing up in [Li et al. 2018], [Ge et al. 2016] and [Chi et al. 2019].
>
> References:
>
> [Hastie et al. 2015] Hastie, T., Tibshirani, R., & Wainwright, M. (2015). Statistical learning with sparsity: the lasso and generalizations. CRC press.
>
> [Li et al. 2018] Li, Y., Ma, T., & Zhang, H. (2018). Algorithmic regularization in over-parameterized matrix sensing and neural networks with quadratic activations. In Conference On Learning Theory (pp. 2-47). PMLR.
>
> [Ge et al. 2016] Ge, R., Lee, J. D., & Ma, T. (2016). Matrix completion has no spurious local minimum. In Advances in Neural Information Processing Systems (pp. 2973-2981).
>
> [Chi et al. 2019] Chi, Y., Lu, Y. M., & Chen, Y. (2019). Nonconvex optimization meets low-rank matrix factorization: An overview. IEEE Transactions on Signal Processing, 67(20), 5239-5269.
>
> [Vaskevicius et al. 2019] Vaskevicius, T., Kanade, V., & Rebeschini, P. (2019). Implicit regularization for optimal sparse recovery. In Advances in Neural Information Processing Systems.

---

### Official Review · AnonReviewer4 · 2020-10-30
**Rigorous results but on a limited problem**

**Rating:** 6
**Confidence:** 4

**Review:**

This paper demonstrates that for a particular model SGD with label noise and proper learning rate schedule recovers the (sparse) data generating model while GD with or without Gaussian noise does not. In the latter case, it fails because a stationary distribution is not achieved. The proofs in the appendix are quite involved, and as a result I did not carefully study them. But the authors provide helpful intuition for the results in the main text.

While I think there is value in the work, I am not sure whether the fairly specific setting studied has much to say about neural networks. (Of course, not every paper needs to be about neural networks, but that's certainly the motivation of the paper.) There are a number of aspects to the work that limit its generality: the model, the label noise, the objective of reconstruction of sparse ground truth from the same model class, and the dataset modeling. The authors justify the model by saying that other works that have studied it, but don't otherwise try to justify its relevance. In Figure 1, the authors argue that label noise behaves similar to SGD, but I don't find this thorough or convincing enough that any results on label noise and the mechanism by which it operates should generalize to SGD. Also, whether studying an objective of learning as identifying the sparse ground truth model seems far from standard training of a neural network. As an example of my concern that the specifics by which the results are achieved may not apply in other scenarios, do the authors believe that GD with Gaussian noise fails for neural networks because a stationary distribution is never achieved, whereas it is achieved for SGD?

Overall, I think that the work is a bit too narrow and doesn't change our understanding of what non-spherical noise from SGD does to neural networks beyond what is already known.


Minor presentation points:

The paper became rushed at the end, as if the authors ran out of space. Subsections 3.2 "Stage 0" and 3.3 "Stage 1" are then followed by a very short paragraph inside 3.3 titled "Stage 2".

Figure 1 is fairly difficult to parse with the number of noisy curves overlapping each other. Perhaps the authors could make their point with the minimal amount of experimental data here and relegate the rest to the appendix.

---

> ### Author Response · Authors · 2020-11-15
> **Response (continued)**
>
> Responses to specific points below:
>
> *-- “Do the authors believe that GD with Gaussian noise fails for neural networks because a stationary distribution is never achieved?”*
>
> Yes, we believe that failing to converge to a good distribution is the reason why GD with Gaussian noise fails. We provide additional empirical results in Section A.3 that can corroborate the hypothesis. We plot the norm of the weight for a VGG19 network trained on CIFAR100 (Figure 2). Our result shows that the weight norm of GD with Gaussian noise keeps increasing and doesn’t converge. In contrast, the weight norm of both GD without noise and GD with label noise converges (with the same decaying learning rate schedule as the Gaussian noise experiment).
>
> References:
>
> [Gunasekar et al. NeurIPS 2017] Gunasekar, S., Woodworth, B., Bhojanapalli, S., Neyshabur, B., & Srebro, N. (2017). Implicit Regularization in Matrix Factorization. In Advances in Neural Information Processing Systems.
>
> [Soudry et al. NeurIPS 2018] Soudry, D., Hoffer, E., Nacson, M. S., Gunasekar, S., & Srebro, N. (2018). The implicit bias of gradient descent on separable data. The Journal of Machine Learning Research, 19(1), 2822-2878.
>
> [Li et al. COLT 2018] Li, Y., Ma, T., & Zhang, H. (2018). Algorithmic regularization in over-parameterized matrix sensing and neural networks with quadratic activations. In Conference On Learning Theory (pp. 2-47). PMLR.
>
> [Vaskevicius et al. NeurIPS 2019] Vaskevicius, T., Kanade, V., & Rebeschini, P. (2019). Implicit regularization for optimal sparse recovery. In Advances in Neural Information Processing Systems.
>
> [Woodworth et al. COLT 2020] Woodworth, B., Gunasekar, S., Lee, J.D., Moroshko, E., Savarese, P., Golan, I., Soudry, D., & Srebro, N.. (2020). Kernel and Rich Regimes in Overparametrized Models. Proceedings of Thirty Third Conference on Learning Theory, in PMLR.
>
> [Wei et al. 2019] Wei, C., Lee, J. D., Liu, Q., & Ma, T. (2019). Regularization matters: Generalization and optimization of neural nets vs their induced kernel. In Advances in Neural Information Processing Systems.
>
> [Ghorbani et al. 2019]Ghorbani, B., Mei, S., Misiakiewicz, T., & Montanari, A. (2019). Limitations of lazy training of two-layers neural networks. In Advances in Neural Information Processing Systems.
>
> [Allen-Zhu et al. 2019] Allen-Zhu, Z., & Li, Y. (2019). What Can ResNet Learn Efficiently, Going Beyond Kernels?. In Advances in Neural Information Processing Systems (pp. 9017-9028).

---

> ### Author Response · Authors · 2020-11-15
> **Response**
>
> We thank the reviewer for the comments and feedback! We will incorporate the minor comments into the next version of our paper and address the major ones below.
>
> The main concern of the reviewer is whether “the fairly specific setting studied has much to say about neural networks”. While we agree with the reviewer that there is a bit gap between our simplified model and deep nets, we would like to clarify that the goal of the work is not to study the mechanisms of implicit regularization in deep neural nets. Instead, our goal is to develop rigorous techniques and understanding for how the noise in optimization can improve the generalization at all in *any* reasonable machine learning setting. Prior to our work, no paper has shown a rigorous result on the improved sample complexity on any problems (including linear models) caused by the noise covariance, and we are the first paper that rigorously distinguishes the effect of spherical Gaussian noise and other non-spherical Gaussian noise.
>
> In our opinion, the paper perhaps should not be evaluated based on the improvement in “our understanding of what non-spherical noise from SGD does in deep learning”, to which the paper does not claim to contribute. Instead, we believe our paper can be better and more fairly evaluated based on the theoretical contributions to proving rigorously that non-spherical Gaussian noise can have a superior implicit regularization effect on some machine learning models, according to which we think we made significant contributions compared to prior work.
>
> For example, the pioneering works [Gunasekar et al. NeurIPS 2017 , Soudry et al. NeurIPS 2018, Li et al., COLT 2018] studies gradient descent’s implicit bias for linear models or low-rank matrix factorization. [Woodworth et al. COLT 2020, Vaskevicius et al. NeurIPS 2019] studied the implicit bias of initialization on the same model as ours. As far as we know, these are all the major rigorous studies of implicit regularization with sample complexity guarantees (please see our related work section for a few other works) and they all work with simpler settings than deep neural nets. These works did not address why stochastic gradient descent with different noises presents different implicit biases, and our work proves it in the same setting as [Woodworth et al. COLT 2020, Vaskevicius et al. NeurIPS 2019].
>
> Arguably, developing rigorous theory is important and analyzing simplified models is an almost necessary first step toward developing rigorous theory for more complex models. It seems that analyzing even GD for deep neural nets is out of reach with our current mathematical techniques (unless we use the NTK approach which results in poor generalization performance as shown by many recent works [Wei et al. 2019, Allen-Zhu et al. 2019, Ghorbani et al. 2019]), let alone analyzing implicit regularization effects of the optimizers.
>
> Therefore, we respectfully ask the reviewers to calibrate our paper against the theoretical works on implicit regularization. (We summarize prior works in the related work section, and would be happy to follow up on any comparisons.) That said, we will also conduct additional experiments on answering the reviewer’s question regarding the relationship to the realistic setting below, though we do not think that’s our main contribution.

---

### Official Review · AnonReviewer2 · 2020-10-31
**Weak accept**

**Rating:** 6
**Confidence:** 3

**Review:**

### Problem

This paper considers the effect of label noise on stochastic gradient descent. The setup is that there is a vector $v \in R^d$. We observe samples from $v^2\cdot x$. We only have $n < d$ samples but $v$ is $r$-sparse for $r < n < d$ which makes recovery possible information theoretically. The main result is that stochastic gradient descent with label noise, and without any explicit regularization will recover the ground truth. whereas adding spherical Gaussian noise does not.

### Pros and Cons

The problem is a clean toy problem with which to illustrate the gap between algorithms. It shows a clean separation between the power of label noise and that of random Gaussian noise. The model appears to be the simplest model where one can hope to see the regularization effects of noise (the simpler linear regression model wouldnt show these effects).  One possible criticism could be to ask if understanding this model is truly getting us closer to understanding what happens in deep nets. At this point it is hard to say, but proving such a result even in this simple model is not trivial, and is definitely a contribution.


### Evaluation
I think this is a solid theoretical contribution on an important problem, and the paper should be accepted.


### Further comments

I was a little confused by the comment that the coefficients are assumed to be in ${0,1}$ since they then satisfy $v_i^2 = v_i$ as this seems to linearize the model. The authors should probably clarify that this is actually not what is going on. It might be better to use a different setting of parameters even for exposition.

---

> ### Author Response · Authors · 2020-11-15
> **Response**
>
> We thank the reviewer for the comments and feedback!
>
> The reviewer’s main concern is that because we assume each entry of the ground truth $v^*$ lies in {0, 1}, “it seems to make the model essentially linear”.  We’d like to clarify that this assumption is only made for simplicity of the exposition--- without much modification of the proof,  we can instead assume that each entry of $v^*$ is either 0 or from [c,C] for constants 0<c<C.
>
> (For instance, to generalize our Theorem 3.1 to the setting where different dimensions of $v^*$ have different values, we only need to change the learning rate accordingly by a constant factor, such that the contraction to 0 still holds. Similarly, we can modify Theorem 3.2 and Theorem 3.3 without changing much.)
>
> We also would like to clarify that the model is *nonlinear* in the trainable parameters. We also don’t make any restriction on whether the trainable parameters are sparse or in {0,1} (or in {0} and [c,C] in the extension). As a consequence, the objective function is non-convex in training parameters. Moreover, we note that the phenomenon of the implicit bias of the noise covariance cannot be empirically observed in standard linear regression models.
>
> Given that we believe that we address the only concern of the reviewer, we respectfully ask the reviewer to consider increasing the score. Our main contribution is a rigorous theory for the quadratically-parameterized model that shows SGD with label noise recovers the sparse ground-truth with an arbitrary initialization, whereas SGD with Gaussian noise or gradient descent overfits and learns dense solutions with large norms. Our analysis reveals that parameter-dependent noise introduces a bias towards local minima with smaller noise variance, whereas spherical Gaussian noise does not.

---

### Decision · Program_Chairs · 2021-01-07
**Final Decision**

**Decision:**

Reject

**Comment:**

The paper shows that for a simple nonlinear (quadratically parametrized linear) model, stochastic gradient descent (SGD) with a certain label noise and learning rate schedule recovers the data generating model. In contrast, gradient descent with or without Gaussian noise fails. While the results are novel and interesting, they hold for a rather specialized model, which may not reveal anything about deep neural networks, which was the original motivation for this work. Given the narrow focus of the work, unfortunately, I cannot recommend that the paper be accepted.